# An Analysis of Causal Effect Estimation using Outcome Invariant Data Augmentation

**Uzair Akbar**[*]
Georgia Tech

**Niki Kilbertus**
TU Munich
Helmholtz AI

**Hao Shen**
TU Munich
Fortiss GmbH

**Krikamol Muandet**
Rational Intelligence
CISPA

**Bo Dai**
Georgia Tech
Google DeepMind

## Abstract

The technique of data augmentation (DA) is often used in machine learning for regularization purposes to better generalize under i.i.d. settings. In this work, we present a unifying framework with topics in causal inference to make a case for the use of DA beyond just the i.i.d. setting, but for generalization across interventions as well. Specifically, we argue that when the outcome generating mechanism is invariant to our choice of DA, then such augmentations can effectively be thought of as interventions on the treatment generating mechanism itself. This can potentially help to reduce bias in causal effect estimation arising from hidden confounders. In the presence of such unobserved confounding we typically make use of instrumental variables (IVs)—sources of treatment randomization that are conditionally independent of the outcome. However, IVs may not be as readily available as DA for many applications, which is the main motivation behind this work. By appropriately regularizing IV based estimators, we introduce the concept of *IV-like (IVL)* regression for mitigating confounding bias and improving predictive performance across interventions even when certain IV properties are relaxed. Finally, we cast parameterized DA as an IVL regression problem and show that when used in composition can simulate a worst-case application of such DA, further improving performance on causal estimation and generalization tasks beyond what simple DA may offer. This is shown both theoretically for the population case and via simulation experiments for the finite sample case using a simple linear example. We also present real data experiments to support our case.

## 1 Introduction

A classical problem in machine learning is that of regression—using i.i.d. samples from some fixed, unknown distribution $\mathbb{P}_{X,Y}$, we predict outcome $Y$ values for unlabelled treatment $X$ values. The use of *regularization* techniques is crucial for this task to achieve good generalization from training to test data [1]. *Data augmentation (DA)* [2, 3] is one such method, where each sample is randomly perturbed multiple times to grow the dataset size. However, these regression models cannot generally be interpreted causally as the statistical relationship between $X$ and $Y$ may arise from shared common causes, known as *confounders*, rather than from $X$ influencing $Y$. Removing such confounders requires independently assigning values of $X$ during data generation, known as an *intervention* [4, 5].

Unfortunately, we seldom have access to the data generation process to be able to intervene on variables. A common workaround is to use auxiliary variables to correct for unobserved confounders [6–8]. One such approach is that of *instrumental variables (IVs)* that represent certain conditional independences in the system which can be used to identify the causal effect of $X$ on $Y$ [9–11]. Alas, IVs too are generally hard to find in may popular applications such as computer vision and natural language processing, motivating the need for more accessible ways to mitigate unobserved confounding.

---

[*]Part of work done while at Max Planck Institute for Intelligent Systems and TU Munich.
This work is a non-archival copy of our NeurIPS 2025 paper [81], for discussion at the Reliable ML Workshop.

39th Conference on Neural Information Processing Systems (NeurIPS 2025).

Recent work therefore seeks to leverage more commonly available auxiliary variables to reduce confounding-induced bias even when the causal effect itself cannot be identified [12–15]. Collectively referred to as *causal regularization*, these methods aim to learn predictors that generalize *out-of-distribution (OOD)* by discouraging reliance on spurious (i.e., non-causal,) correlations. Since distribution shifts often correspond to interventions on parts of the data-generating process [16, 4], models that fail under such shifts typically do so because they exploit confounded relationships [17]. Tackling this root cause directly, causal regularization offers a principled approach for more robust prediction.

In the same vein, more ambitious works have also explored the use of common regularization techniques, such as $\ell_1$, $\ell_2$ [18] and the min-norm interpolator [19], for the same purpose of causal regularization. This is in contrast to the canonical use of such regularizers for estimation variance reduction and i.i.d. prediction generalization [1]. Other popular regularization methods, however, remain understudied in a similar context of un-identifiable causal effect estimation, motivating our work.

**Our contributions.** To this end, we provide a first analysis of DA for estimating un-identifiable causal effects using only observational data for $(X, Y)$. Our contributions, summarized in Tab. 1, include: (i) **DA as a soft intervention (Sec. 4.1):** We show that DA can synthesize treatment interventions when the outcome function is invariant to DA, lowering bias in causal effect estimates when the intervention acts along spurious features. (ii) **Introducing IV-like regression (Sec. 3):** Relaxing the properties of IVs, we introduce the concept of *IV-like (IVL)* variables. This generalization renders IV regression ineffective at identifying causal effects, but when regularized appropriately via our proposed *IVL regression*, may still reduce confounding bias and improve prediction generalization across treatment interventions. (iii) **DA parameters as IVL (Sec. 4.2):** By casting parameterized DA as IVL, we show that its composition DA+IVL with IVL regression further reduces confounding bias beyond just simple DA by essentially simulating a worst-case or adversarial application of the DA.

We validate our approach with theoretical results in a linear setting for the infinite-sample case, and simulation and real-data experiments in the finite-sample case.

## 2 Preliminaries

Consider treatment $X$ and outcome $Y$ taking values in $\mathcal{X} \subseteq \mathbb{R}^m$ and $\mathcal{Y} \subseteq \mathbb{R}^l$ respectively. Given the set of functions $\mathcal{H} := \{h : \mathcal{X} \to \mathcal{Y}\}$, the canonical setting described in the literature [4, 15, 20] deals with estimating the function $f \in \mathcal{H}$ in the *structural equation model (SEM)* $\mathfrak{M}$ of the following form[1]

$$X = \tau(Y, Z, C, N_X), \qquad\qquad Y = f(X) + \epsilon(C) + N_Y, \qquad\qquad (1)$$

where $Z, C, N_X, N_Y$ are exogenous (and therefore mutually independent) random variables and the residual $\xi := Y - f(X) = \epsilon(C) + N_Y$ is assumed to be zero mean, i.e. $\mathbb{E}^{\mathfrak{M}}[\xi] = 0$. Since $\mathfrak{M}$ is potentially cyclic, a priori it may entail several or no distributions at all. However, here we make the assumption that for all $(\mathbf{x}_0, \mathbf{y}_0) \in \mathcal{X} \times \mathcal{Y}$ the unique limits

$$\mathbf{x} := \lim_{t\to\infty} \mathbf{x}_t = \lim_{t\to\infty} \tau(\mathbf{y}_{t-1}, \mathbf{z}, \mathbf{c}, \mathbf{n}_X), \qquad \mathbf{y} := \lim_{t\to\infty} \mathbf{y}_t = \lim_{t\to\infty} f(\mathbf{x}_{t-1}) + \epsilon(\mathbf{c}) + \mathbf{n}_Y$$

exist for any $(\mathbf{z}, \mathbf{c}, \mathbf{n}_X, \mathbf{n}_Y) \sim \mathbb{P}^{\mathfrak{M}}_{Z,C,N_X,N_Y}$, meaning that the unique distribution entailed by $\mathfrak{M}$ is in this equilibrium state. Of course, if $\mathfrak{M}$ is acyclic, these limits always exist. Note that assuming the existence of such an equilibrium does not violate the classic *independent causal mechanism (ICM)* principle [4]; we defer interested readers to Appendix B for further details on cyclic SEMs and the ICM.

Given a proper convex loss $\ell : \mathbb{R}^l \times \mathbb{R}^l \to \mathbb{R}_+$, *empirical risk minimization (ERM)* uses a dataset $\mathcal{D} := \{(\mathbf{x}_i, \mathbf{y}_i)\}_{i=0}^{n}$ of $n$ samples from $\mathfrak{M}$ to minimize an empirical version of the *statistical risk*

$$R^{\mathfrak{M}}_{\mathrm{ERM}}(h) := \mathbb{E}^{\mathfrak{M}}[\ell(Y, h(X))], \qquad\qquad (2)$$

over $h \in \mathcal{H}$. However, since the residual $\xi$ in Eq. (1) is generally correlated with $X$, i.e., $\mathbb{E}^{\mathfrak{M}}[\xi \mid X] \neq 0$, the ERM minimizer $\hat{h}^{\mathfrak{M}}_{\mathrm{ERM}}$ typically yields a biased estimate of $f$ [5, 4]. This bias arises due to the exclusion of the (unobserved) common parent $C$ of $X$ and $Y$, i.e. a confounder, in the ERM objective (hence fittingly called the *omitted-variable bias* [21]) and/or the model is cyclic (*simultaneity bias* [20, 22], or *reverse causality* [5] in the degenerate case). For simplicity we shall refer to either case by saying that $X$ and $Y$ are confounded and the resulting bias as the *confounding bias* [5].[2]

---

[1]Throughout this work we shall borrow and overload notation from [4]. See Appendix for a list of symbols.
[2]Pearl [5, p.78,184] similarly uses the term for any bias causing observational vs. interventional deviation; this also aligns with econometrics [23, 20], where both are classified as sources of *endogeneity* (i.e., $X \not\perp\!\!\!\perp \xi$).

Table 1: A picture summary of our contributions. $\rightarrow$ represents composition of operations or transformations, and $\Leftrightarrow$ represents equivalence.

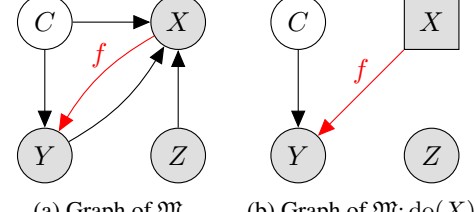

| | Type of Data Augmentation | | Topics in Causal Inference |
|---|---|---|---|
| | None; observational data | $\leftarrow$ | Data generating structural model |
| | $\downarrow$ | | $\downarrow$ |
| | Outcome invariant DA | $\overset{(i)}{\Longleftrightarrow}$ | Treatment (soft) intervention |
| | $\downarrow$ | | $\downarrow$ |
| | Worst-case or adversarial DA | $\overset{(iii)}{\Longleftrightarrow}$ | Regularized IV regression (ii) |

(vertical label at left: *Lower confounding bias in causal effect estimate*)

(a) Graph of $\mathfrak{M}$.  (b) Graph of $\mathfrak{M}; \mathrm{do}(X)$.

Figure 1: Graph of $\mathfrak{M}$ depicting an instrument $Z$ that satisfies treatment relevance, exclusion restriction, un-confoundedness and outcome relevance properties. An intervention on $X$ gives us the graph in *(b)*. IV regression simulates such an intervention using only observational data.

## 2.1 Intervention for causal effect estimation

We can make $X$ and the residual $\xi$ uncorrelated via an intervention[3] $\mathrm{do}(X := X')$, where we explicitly set $X$ to some independently sampled $X'$ in Eq. (1) irrespective of its parents, resulting now in the new SEM $\mathfrak{M}; \mathrm{do}(X := X')$ or $\mathfrak{M}; \mathrm{do}(X)$ as a shorthand for when $X' \sim \mathbb{P}_X^{\mathfrak{M}}$. The distribution induced by this modified SEM is called an *interventional distribution* (with respect to $\mathfrak{M}$) under which the ERM objective from Eq. (2) now defines the following *causal risk (CR)* [12, 19, 24] as

$$R_{\mathrm{CR}}^{\mathfrak{M}}(h) := R_{\mathrm{ERM}}^{\mathfrak{M};\mathrm{do}(X)}(h) = R_{\mathrm{ERM}}^{\mathfrak{M};\mathrm{do}(X := X')}(h), \qquad \text{s.t.} \qquad X' \sim \mathbb{P}_X^{\mathfrak{M}}. \qquad (3)$$

Minimizing Eq. (3) is meaningful in two important cases where ERM fails: (i) **Causal effect estimation:** The minimizer $\hat{h}_{\mathrm{CR}}^{\mathfrak{M}}$ of Eq. (3) gives us an unbiased estimate of the *average treatment effect (ATE)* [6] $\mathbb{E}^{\mathfrak{M};\mathrm{do}(X := \mathbf{x})}[Y \mid X = \mathbf{x}] = f(\mathbf{x})$ that measures the causal influence of $X$ on $Y$. (ii) **Robust prediction:** ATE based prediction of $Y$ values for unlabelled $X$ values is *robust* in the sense that it can generalize across arbitrary OOD treatment interventions or shifts in the treatment distribution [25]. Consequently, the causal risk minimizer $\hat{h}_{\mathrm{CR}}^{\mathfrak{M}}$ is also a robust predictor over the support of $\mathbb{P}_X^{\mathfrak{M}}$. Specifically, $\hat{h}_{\mathrm{CR}}^{\mathfrak{M}}$ minimizes the worst-case ERM objective over the set $\mathcal{P}$ of all possible intervention distributions $\mathbb{P}_{X'}$ over the support of $\mathbb{P}_X^{\mathfrak{M}}$ [25], i.e. for $\mathcal{P} := \{\mathbb{P}_{X'} \mid \mathrm{supp}(\mathbb{P}_{X'}) \subseteq \mathrm{supp}(\mathbb{P}_X^{\mathfrak{M}})\}$,

$$\hat{h}_{\mathrm{CR}}^{\mathfrak{M}} \in \underset{h \in \mathcal{H}}{\mathrm{argmin}} \max_{\mathbb{P}_{X'} \in \mathcal{P}} R_{\mathrm{ERM}}^{\mathfrak{M};\mathrm{do}(X := X')}(h).$$

To better isolate the estimation error due to confounding, we define the *causal excess risk (CER)* [19]

$$\mathrm{CER}_{\mathfrak{M}}(h) := R_{\mathrm{CR}}^{\mathfrak{M}}(h) - R_{\mathrm{CR}}^{\mathfrak{M}}(f).$$

This removes the irreducible noise from Eq. (3) (see Appendix A) and directly measures how far a hypothesis $h$ deviates from the true causal function $f$ under interventions, so that $\mathrm{CER}_{\mathfrak{M}}(f) = 0$.

Since interventions are often inaccessible for computing the risk in Eq. (3), we usually rely on observational data/ distribution and additional variables to approximate them, as outlined in the next section.

## 2.2 Instrumental variable regression

One way to get an unbiased estimate of $f$ from the observational distribution of $\mathfrak{M}$ is to use so-called instrumental variables $Z$ with the properties [5, 4, 10, 9, 26] of: (i) **Treatment Relevance:** $Z \not\perp\!\!\!\perp X$. (ii) **Exclusion Restriction:** $Z$ enters $Y$ only through $X$, i.e. $Z \perp\!\!\!\perp Y^{\mathfrak{M};\mathrm{do}(X:=\mathbf{x})}$.[4] (iii) **Un-confoundedness:** $Z \perp\!\!\!\perp \xi$. (iv) **Outcome Relevance:** $Z$ carries information about $Y$, i.e. $Y \not\perp\!\!\!\perp Z$.

---

[3] A *soft* intervention replaces the mechanism $\tau$ in Eq. (1) with an alternative $\tau'$ [4, p. 34]. This may *potentially* reduce confounding between $X$ and $Y$.

[4] Counterfactual definition of the exclusion restriction property [5, p. 248].

Conditioning Eq. (1) on $Z$ and using $\mathbb{E}[\xi \mid Z] = \mathbb{E}[\xi] = 0$ from the unconfoundedness property gives

$$\mathbb{E}^{\mathfrak{M}}[Y \mid Z] = \mathbb{E}^{\mathfrak{M}}[f(X) \mid Z]. \tag{4}$$

IV regression therefore entails solving Eq. (4) for $f$, which can be done by minimizing the risk [26]

$$R_{\mathrm{IV}}^{\mathfrak{M}}(h) := \mathbb{E}^{\mathfrak{M}}\big[\ell\big(Y, \mathbb{E}^{\mathfrak{M}}[h(X) \mid Z]\big)\big]. \tag{5}$$

For linear $f(\cdot) := \mathbf{f}^\top(\cdot), h(\cdot) := \mathbf{h}^\top(\cdot)$ with $\mathbf{f}, \mathbf{h} \in \mathbb{R}^m$ and squared loss $\ell(\mathbf{y}, \mathbf{y}') := \|\mathbf{y} - \mathbf{y}'\|^2$, this gives the two-stage-least-squares (2SLS) [27] solution where the first stage regresses $X$ from $Z$, and the second stage regresses $Y$ from predictions $\mathbb{E}[X \mid Z]$ of the first stage to get the estimate $\hat{h}_{\mathrm{IV}}^{\mathfrak{M}}$.

## 2.3 Data augmentation

In this work we restrict ourselves to data augmentation with respect to which $f$ is invariant [3, 28]. The action of a group $\mathcal{G}$ is a mapping $\delta : \mathcal{X} \times \mathcal{G} \to \mathcal{X}$ which is compatible with the group operation. For convenience we shall write $\mathbf{gx} := \delta(\mathbf{x}, \mathbf{g})$. We say that $f$ is *invariant* under $\mathcal{G}$ (or $\mathcal{G}$-*invariant*) if

$$f(\mathbf{gx}) = f(\mathbf{x}), \qquad \forall\, (\mathbf{g}, \mathbf{x}) \in \mathcal{G} \times \mathcal{X}.$$

Less formally, we say that the map $\mathbf{gx}$, henceforth assumed to be continuous in $\mathbf{x}$, is a valid *outcome-invariant* DA transformation parameterized by the vector $\mathbf{g} \in \mathcal{G}$. Let $\mathcal{G}$ have a (unique) normalized Haar measure and $\mathbb{P}_G$ be the corresponding distribution defined over it. For some $G \sim \mathbb{P}_G$, the canonical application of DA seeks to minimize an empirical version of the following risk.

$$R_{\mathrm{DA}_G+\mathrm{ERM}}^{\mathfrak{M}}(h) := \mathbb{E}^{\mathfrak{M}}[\ell(Y, h(GX))]. \tag{6}$$

Note that it is sufficient to have some prior information about the symmetries of $f$ in order to be able to construct such a DA. For example, when classifying images of cats and dogs we already know that whatever the true labeling function may be, it would certainly be invariant to rotations on the images. $G$ would then represent the random rotation angle, whereas $G\mathbf{x}$ would be the rotated image $\mathbf{x}$.

We wish to contrast the use of DA in this work with the canonical setting—to mitigate overfitting, DA is used to grow the sample size by generating multiple augmentations $(G\mathbf{x}, \mathbf{y})$ for each data sample $(\mathbf{x}, \mathbf{y}) \sim \mathbb{P}_{X,Y}^{\mathfrak{M}}$ [3, 28, 29]. Such regularization, overfitting mitigation, estimation variance reduction, or i.i.d. prediction generalization is not the focus of this work and we intentionally provide Eq. (6) along with theoretical results that follow in the population case to emphasize that DA is not being used as a conventional regularizer. Instead, our goal is to improve causal effect estimation and robust prediction by re-purposing DA to mitigate hidden confounding bias in the data.

## 3 Faithfulness and Outcome Relevance in IVs

The distribution $\mathbb{P}_{X,Y,Z,C}^{\mathfrak{M}}$ is *faithful* to the graph of $\mathfrak{M}$ if it only exhibits independences implied by the graph [4, 30].[5] This standard assumption in IV settings renders outcome-relevance implicit and therefore rarely mentioned. In this section we discuss the case where only the first three IV properties are satisfied, i.e. outcome-relevance may not hold. Since such a $Z$ may not be a valid IV, therefore identifiability of ATE is not possible in general as the problem in Eq. (4) can now be misspecified, having multiple, potentially infinitely many solutions when $Y \perp\!\!\!\perp Z$. Nevertheless, we shall refer to such a $Z$ as *IV-like (IVL)* to emphasize that while $Z$ may not be an IV, it may still be 'instrumental' for reducing confounding bias when estimating the ATE compared to the standard ERM baseline.

**ERM regularized IV regression.** Despite problem miss-specification for a IVL $Z$, the target function $f$ remains a minimizer for the IV risk in Eq. (5). Albeit, potentially not unique—for example, a linear $h$ with squared loss leads to an under-determined problem in Eq. (5). We therefore propose the following regularized version of the IV risk for such an IVL setting,

$$R_{\mathrm{IVL}_\alpha}^{\mathfrak{M}}(h) := R_{\mathrm{IV}}^{\mathfrak{M}}(h) + \alpha R_{\mathrm{ERM}}^{\mathfrak{M}}(h), \tag{7}$$

where $\alpha > 0$ is the regularization parameter. The ERM risk as a penalty allows our estimations to have good predictive performance while the IV risk encourages solution search within the subspace where we know $f$ to be present. We refer to minimising the risk in Eq. (7) as *IVL regression*.

Note that the motivation behind IVL regression is not the identifiability of $f$, but rather potentially better estimations of $f$ with lower confounding bias. The next section provides a concrete example.

---

[5]Also known as *stability* in some texts [5, p. 48].

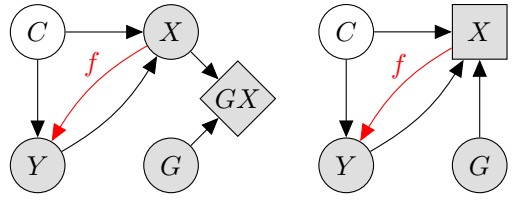

(a) Graph of $\mathfrak{A}$ post DA.   (b) Graph of $\mathfrak{A}; \mathrm{do}(\tau := G\tau)$

Figure 2: The observational distribution of $(GX, Y, G, C)$ and $(X, Y, G, C)$ for graphs *(a)* and *(b)* respectively are the same. The former applies DA on $X$, whereas the later applies a (soft) intervention on $X$. Furthermore, for the graph in *(b)*, $G$ is IVL.

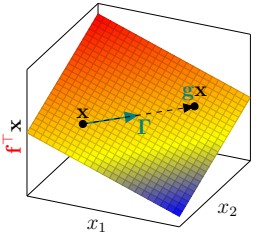

Figure 3: The ground truth function $\mathbf{f}$ in Example 2. The DA applied here corresponds to randomly translating the data samples along their level-set by adding random noise sampled from the null-space of $\mathbf{f}$.

**Example 1** (a linear Gaussian IVL example). For scalar $\sigma > 0$, non-zero matrices $\mathbf{\Gamma}, \mathbf{T} \in \mathbb{R}^{* \times m}$ and vectors $\boldsymbol{\tau}^\top, \mathbf{f}, \boldsymbol{\epsilon} \in \mathbb{R}^m$ such that $\mathbf{f}^\top \boldsymbol{\tau}^\top \neq 1$ so that the following SEM $\mathfrak{M}$ is solvable in $(X, Y)$[6]

$$X = \boldsymbol{\tau}^\top Y + \mathbf{\Gamma}^\top Z + \mathbf{T}^\top C + \sigma N_X, \qquad Y = \mathbf{f}^\top X + \boldsymbol{\epsilon}^\top C + \sigma N_Y,$$

where $Z, C, N_X, N_Y$ are conformable, centered Gaussian random vectors and $Z$ is IVL w.r.t. $(X, Y)$.[7]

Now, the task is to improve our estimation of $\mathbf{f}$ compared to standard ERM. We evaluate an estimate $\hat{\mathbf{h}}^{\mathcal{D}}$ using the CER, which for a squared loss and covariance $\mathbf{\Sigma}_X^{\mathfrak{M}}$ in Example 1 simply comes out to be

$$\mathrm{CER}_{\mathfrak{M}}\left(\hat{\mathbf{h}}^{\mathcal{D}}\right) = \left\|\hat{\mathbf{h}}^{\mathcal{D}} - \mathbf{f}\right\|_{\mathbf{\Sigma}_X^{\mathfrak{M}}}^2. \tag{8}$$

Prior works use this form to quantify the error in ATE estimation [19, 12] or measure some notion of strength of confounding [18, 31, 24]. Similarly, we use it to measure confounding bias of population estimates $\hat{\mathbf{h}}^{\mathfrak{M}}$ (Appendix A) and estimation error in finite sample experiments. The next results follow.

**Theorem 1** (robust prediction with IVL regression). *For SEM $\mathfrak{M}$ in Example 1, the following holds:*

$$\hat{\mathbf{h}}_{IVL_\alpha}^{\mathfrak{M}} \in \underset{\mathbf{h}}{\mathrm{argmin}} \max_{\boldsymbol{\zeta} \in \mathcal{P}_\alpha} R_{ERM}^{\mathfrak{M}; \mathrm{do}(\mathbf{\Gamma}^\top(\cdot) := \boldsymbol{\zeta})}(\mathbf{h}), \quad s.t. \quad \mathcal{P}_\alpha := \left\{\boldsymbol{\zeta} \mid \boldsymbol{\zeta}\boldsymbol{\zeta}^\top \preccurlyeq \left(\frac{1}{\alpha} + 1\right)\mathbf{\Gamma}^\top \mathbf{\Sigma}_Z^{\mathfrak{M}} \mathbf{\Gamma}\right\}.$$

*Proof.* See Appendix F.3 for the proof. □

**Theorem 2** (causal estimation with IVL regression). *In SEM $\mathfrak{M}$ of Example 1, for $\alpha < \infty$, we have*

$$\mathrm{CER}_{\mathfrak{M}}\left(\hat{\mathbf{h}}_{IVL_\alpha}^{\mathfrak{M}}\right) \leq \mathrm{CER}_{\mathfrak{M}}\left(\hat{\mathbf{h}}_{ERM}^{\mathfrak{M}}\right), \qquad \textit{equality iff} \qquad \mathbb{E}^{\mathfrak{M}}[X \mid Z] \perp_{\mathrm{a.s.}} \mathbb{E}^{\mathfrak{M}}[X \mid \xi].$$

*Proof.* See Appendix F.4 for the proof. □

Theorem 1 shows that IVL regression achieves optimal predictive performance across treatment interventions within the perturbation set $\mathcal{P}_\alpha$ defined by $\alpha$. Theorem 2 further states that this strictly reduces confounding bias in ATE estimates iff the perturbations align with spurious features of $X$, as indicated by the equality condition (also necessary for identifiability in linear IV settings [32, 25]).

## 4   Causal Effect Estimation using Data Augmentation

We dedicate this section to the main topic and point of this work—discussing the potential of data augmentation for improving predictive performance across interventions and reducing confounding bias in ATE estimates. To that effect, for the rest of this work we shall consider the following SEM $\mathfrak{A}$

$$X = \tau(Y, C, N_X), \qquad Y = f(X) + \epsilon(C) + N_Y, \tag{9}$$

which is assumed to have a unique stationary distribution with exogenous $C, N_X, N_Y$ and the residual $\xi := Y - f(X)$ is zero-mean, i.e. $\mathbb{E}[\xi] = 0$. We also have access to DA transformations $GX$ of $X$ parameterized by $G \sim \mathbb{P}_G^{\mathfrak{A}}$ such as described in Sec. 2.3. Figure 2a shows the graph of $\mathfrak{A}$ post DA.

Given samples for only $(X, Y)$ and some valid DA parameterised by $G$, the task is to improve predictive performance across interventions and reduce confounding bias in ATE estimates. We now make two observations in the following sections and state the respective results that follow thereof.

---

[6]See Appendix B and Lemma 3 for details on solving for and sampling of $(X, Y)$ in such linear, cyclic SEMs.
[7]All examples assume correlated $X$ and residual $\xi$, i.e. $\mathbb{E}^{\mathfrak{M}}\left[X\xi^\top\right] \neq \mathbf{0}$, as otherwise there is no confounding.

### 4.1 Data augmentation as a soft intervention

Consider a (soft) intervention on $\mathfrak{A}$ where we substitute the mechanism $\tau$ of $X$ with $G\tau$. With some abuse of notation, we shall represent this SEM by $\mathfrak{A}; \mathrm{do}(\tau := G\tau)$ the graph of which is shown in Fig. 2b. Note that this SEM also has a unique stationary distribution (proof in Appendix F.2). Comparing the DA mechanism in $\mathfrak{A}$ (Fig. 2a) and the intervention $\mathfrak{A}; \mathrm{do}(\tau := G\tau)$ (Fig. 2b), we see:

**Observation 1** (soft intervention with DA). *Distributions* $\mathbb{P}^{\mathfrak{A}}_{GX,Y,G,C}$ *and* $\mathbb{P}^{\mathfrak{A};\mathrm{do}(\tau:=G\tau)}_{X,Y,G,C}$ *are identical.*

We can hence treat samples generated from $\mathfrak{A}$ via DA as if they were instead generated from $\mathfrak{A}; \mathrm{do}(\tau := G\tau)$ by intervening on $X$. This allows us to re-write the DA+ERM risk from Eq. (6) as,

$$R^{\mathfrak{A}}_{\mathrm{DA}_G+\mathrm{ERM}}(h) = R^{\mathfrak{A};\mathrm{do}(\tau:=G\tau)}_{\mathrm{ERM}}(h),$$

to emphasize that DA is equivalent to a (soft) intervention and as such can be used to reduce confounding bias when estimating $f$, as we will show with the following example.

**Example 2** (a linear Gaussian DA example). *For scalars* $\kappa, \sigma > 0$, *non-zero matrices* $\mathbf{\Gamma}, \mathbf{T} \in \mathbb{R}^{*\times m}$ *and vectors* $\boldsymbol{\tau}^\top, \mathbf{f}, \boldsymbol{\epsilon} \in \mathbb{R}^m$ *such that* $\mathbf{f}^\top \boldsymbol{\tau}^\top \neq \kappa^{-1}$ *so that the following SEM* $\mathfrak{A}$ *is solvable in* $(X, Y)$

$$X = \kappa \cdot \boldsymbol{\tau}^\top Y + \mathbf{T}^\top C + \sigma N_X, \quad Y = \mathbf{f}^\top X + \kappa \cdot \boldsymbol{\epsilon}^\top C + \sigma N_Y, \quad GX := X + \gamma \cdot \mathbf{\Gamma}^\top G,$$

*where* $G, C, N_X, N_Y$ *are conformable, centered Gaussian random vectors,* $\kappa$ *determines how much* $(X, Y)$ *are confounded and* $\mathrm{range}(\mathbf{\Gamma}^\top) \subseteq \mathrm{null}(\mathbf{f}^\top)$ *so that* $GX$ *is a valid outcome invariant DA transformation of* $X$ *parameterized by* $G$ *with* strength $\gamma > 0$. *This transformation can be viewed as translating* $X$ *along its level-set as shown in Fig. 3 and represents our prior knowledge about the symmetries of* $\mathbf{f}$ *for the purposes of this example.*

**Theorem 3** (causal estimation with DA+ERM). *For SEM* $\mathfrak{A}$ *in Example 2, the following holds:*

$$\mathrm{CER}_{\mathfrak{A}}\left(\hat{\mathbf{h}}^{\mathfrak{A}}_{DA_G+ERM}\right) \leq \mathrm{CER}_{\mathfrak{A}}\left(\hat{\mathbf{h}}^{\mathfrak{A}}_{ERM}\right), \quad \textit{equality iff} \quad \mathbb{E}^{\mathfrak{A}}[GX \mid G] \perp_{\mathrm{a.s.}} \mathbb{E}^{\mathfrak{A}}[X \mid \xi].$$

*Proof.* See Appendix F.5 for the proof. $\qquad\square$

That is, DA strictly reduces confounding bias in ATE estimate iff the induced intervention perturbes $X$ along spurious features. Importantly, Theorem 3 suggests that lower confounding bias is not a 'free lunch' with outcome invariance of DA and practitioners may need domain knowledge to construct DA that targets spurious features. Fortunately however, Theorem 3 also suggests that with outcome invariance, DA should not perform worse than ERM. We say that DA+ERM *dominates* ERM on causal estimation [33, p. 48]. Practitioners may therefore be advised to generously use such DA, as it achieves regularization in the worst case, and mitigates confounding bias as a 'bonus' in the best case.

### 4.2 Worst-case data augmentation with IVL regression

We once again point our attention to the graph of $\mathfrak{A}; \mathrm{do}(\tau := G\tau)$ from Fig. 2b to observe that:

**Observation 2** (IV-like DA parameters). *In SEM* $\mathfrak{A}; \mathrm{do}(\tau := G\tau)$, *the DA parameters* $G$ *are IVL.*

In light of this we can now re-write the IV and IVL risks for $\mathfrak{A}; \mathrm{do}(\tau := G\tau)$ to respectively read

$$R^{\mathfrak{A}}_{\mathrm{DA}_G+\mathrm{IV}}(h) = R^{\mathfrak{A};\mathrm{do}(\tau:=G\tau)}_{\mathrm{IV}}(h), \qquad R^{\mathfrak{A}}_{\mathrm{DA}_G+\mathrm{IVL}_\alpha}(h) = R^{\mathfrak{A};\mathrm{do}(\tau:=G\tau)}_{\mathrm{IVL}_\alpha}(h).$$

**Corollary 1** (worst-case DA with DA+IVL regression). *For SEM* $\mathfrak{A}$ *in Example 2, it holds that*

$$\hat{\mathbf{h}}^{\mathfrak{A}}_{DA_G+IVL_\alpha} \in \underset{\mathbf{h}}{\mathrm{argmin}}\, \underset{\mathbf{g}\in\mathcal{G}_\alpha}{\max}\, R^{\mathfrak{A}}_{DA_\mathbf{g}+ERM}(\mathbf{h}), \quad s.t. \quad \mathcal{G}_\alpha := \left\{\mathbf{g} \,\middle|\, \mathbf{\Gamma}^\top \mathbf{g}\mathbf{g}^\top \mathbf{\Gamma} \preccurlyeq \left(\frac{1}{\alpha}+1\right)\mathbf{\Gamma}^\top \mathbf{\Sigma}^{\mathfrak{A}}_G \mathbf{\Gamma}\right\}.$$

*Proof.* The result follows from Observation 1, Observation 2 and Theorem 1. $\qquad\square$

**Corollary 2** (causal estimation with DA+IVL regression). *For* $\alpha, \gamma < \infty$ *in SEM* $\mathfrak{A}$ *from Example 2,*

$$\mathrm{CER}_{\mathfrak{A}}\left(\hat{\mathbf{h}}^{\mathfrak{A}}_{DA_G+IVL_\alpha}\right) \leq \mathrm{CER}_{\mathfrak{A}}\left(\hat{\mathbf{h}}^{\mathfrak{A}}_{DA_G+ERM}\right), \quad \textit{equality iff} \quad \mathbb{E}^{\mathfrak{A}}[GX \mid G] \perp_{\mathrm{a.s.}} \mathbb{E}^{\mathfrak{A}}[X \mid \xi].$$

*Proof.* The result follows directly from Theorem 2 and Observation 2. $\qquad\square$

Using DA parameters as IVL therefore simulates a worst-case, or adversarial application of DA within a set of transforms $\mathcal{G}_\alpha$. Of course Corollary 1 can also be viewed as a predictor that generalizes to treatment interventions encoded by $\mathcal{G}_\alpha$. As is intuitive, such a worst-case intervention improves our ATE estimation so long as the features of $X$ intervened along include some that are spurious (Corollary 2). DA and IVL regression may therefore be used in composition if the application can benefit from regularization and/ or better prediction generalization across DA-induced interventions, with a 'bonus' of lower confounding bias if the DA also augments any spurious features of $X$.

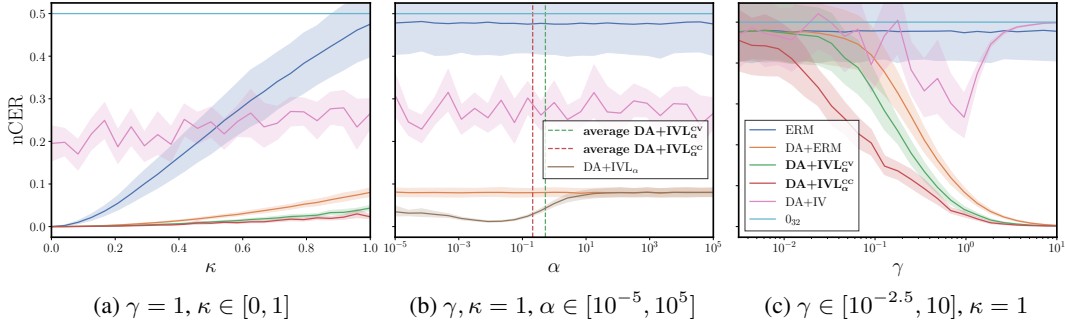

(a) $\gamma = 1, \kappa \in [0, 1]$      (b) $\gamma, \kappa = 1, \alpha \in [10^{-5}, 10^5]$      (c) $\gamma \in [10^{-2.5}, 10], \kappa = 1$

Figure 4: Simulation experiment for a linear Gaussian SEM. $\kappa$ represents the amount of confounding, $\gamma$ is the strength of DA and $\alpha$ is the IVL regularization parameter. Each data-point represents the average nCER over 25 trials with a 95% confidence interval (CI).

## 5 Related Work

**Causal regularization** is perhaps the most appropriate classification for this work. These methods aim for more robust prediction by mitigating the upstream problem of confounding bias in a more accessible way than is required for full identification. This is done, for example, by relaxing properties of auxiliary variables [12–15], as we have done via our IVL approach. Most relevant, however, are methods that re-purpose common regularizers, canonically used for estimation variance reduction and i.i.d. prediction generalization, for confounding bias mitigation. Of note is [18], where a certain linear modeling assumption allows the estimation of $\|\mathbf{f}\|^2$ from observational $(X, Y)$ data, which is then used to develop a cross-validation scheme for $\ell_1, \ell_2$ regularization. [19] conducted a similar theoretical analysis for the min-norm interpolator. To the best of our knowledge, we are the first to study the same for DA—re-purposing yet another ubiquitous regularizer to mitigate confounding bias.

**Domain generalization (DG)** [34] methods aim for prediction generalization to unseen test domains via *robust optimization (RO)* [35] over a perturbation set $\mathcal{P}$ of possible test domains $\rho \in \mathcal{P}$ as

$$R_{\text{RO}}^{\mathcal{P}}(h) := \max_{\rho \in \mathcal{P}} R_{\text{ERM}}^{\rho}(h),$$

Since generalizing to arbitrary test domains is impossible, the choice of perturbation set encodes one's assumptions about which test domains might be encountered. Instead of making such assumptions a priori, it is often assumed to have access to data from multiple training domains which can inform one's choice of perturbation set. This setting is explored in group distributionally robust optimization (DRO) [36]. Variations have been used to mitigate confounding bias and subsequently generalize to treatment interventions when used with interventional data [16, 37], confounder information (i.e. entire graph) [38–40] or some proxy thereof in the form of environments [41–43, 38]. We, however, do not assume access to any of these and instead synthesize interventions via DA.

**Counterfactual DA** strategies have been the primary lens for causal analyses of DA [44–50]. These aim for prediction robustness to treatment interventions via DA simulated *counterfactuals*.[8] As with counterfactual reasoning more broadly, this requires strong assumptions—such as access to the full SEM [45, 46], auxiliary variables [44, 46, 49, 50], or causal graphs [47, 48]. By contrast, we show that outcome invariance of DA suffices for treatment intervention robustness without invoking counterfactuals. Moreover, prior work has largely overlooked causal effect estimation, often assuming reverse-causal settings where the ATE becomes trivial [44, 46, 45]. Ours is the first framework to study ATE estimation via DA with minimal structural assumptions.

**Invariant prediction** based methods aim to make predictions based on statistical relationships that remain stable across all domains in $\mathcal{P}$. A common assumption, for instance, is that $\mathbb{P}_{Y|X}$ is invariant across $\mathcal{P}$, with only the marginal $\mathbb{P}_X$ being allowed to vary. Invariance is also closely linked to causal discovery—following the classic ICM principle [4], causal mechanisms remain stable under interventions on inputs [25, 17]. This connection has inspired approaches that enforce invariance conditions

---

[8]Representing an SEM with exogenous noise distribution conditioned on some variable $Y = \mathbf{y}$ by $\mathfrak{A}_{Y=\mathbf{y}}$, the counterfactual SEM $\mathfrak{A}_{Y=\mathbf{y}}; \text{do}(X := \mathbf{x})$ is an intervention $\text{do}(X := \mathbf{x})$ on $\mathfrak{A}_{Y=\mathbf{y}}$. The resulting *counterfactual distribution* then captures questions like: "After observing $Y = \mathbf{y}$, what would have been had $X = \mathbf{x}$ been true."

to recover causal structures [16, 37]. IV regression can also be viewed as one such method, where the goal is to learn predictors whose residuals are invariant to the instruments [10, 9, 26, 51, 7]. More broadly, the principle of invariance, whether motivated by causality or otherwise, has proven useful for improving prediction generalization across heterogeneous settings [15, 41, 52, 14, 53–56, 34].

## 6 Experiments

We began by presenting results in the infinite-sample setting to emphasize that mitigating confounding bias is fundamentally not a sample size issue, i.e., not solvable through traditional regularization alone. In this section, we turn to the finite-sample regime and empirically evaluate the effectiveness of DA in reducing hidden confounding bias. Importantly, we do not use DA for its conventional purpose of augmenting data to improve i.i.d. generalization or reduce estimation variance. Throughout all experiments, we therefore fix the number of samples in the augmented dataset to match that of the original dataset since our focus lies squarely on robust prediction via confounding bias mitigation.

Finding baselines for evaluating our results is however a challenge—the problem of mitigating confounding bias given only observational $(X, Y)$ data and symmetry knowledge via DA is quite underexplored. Nevertheless, for the sake of completeness we make an effort to re-purpose existing methods from domain generalization, invariance learning and causal inference literature to be used as baselines. These methods often require access to additional variables (e.g. IVs, confounders, domains/environments, etc.), and to maintain fairness we will replace these with DA parameters $G$. Such a comparison is conceptually valid since by virtue of being DG methods, they are essentially solving a robust loss of a similar form as in Corollary 1, giving us meaningful baselines for DA+IVL.

In addition to standard ERM, DA and IV regression, our baselines include DRO [36], invariant risk minimization (IRM) [41], invariant causal prediction (ICP) [16], regularization with invariance on causal essential set (RICE) [56], variance risk extrapolation (V-REx) and minimax risk extrapolation (MM-REx) [38]. We also include the causal regularization method by Kania and Wit [12] and the $\ell_1, \ell_2$ approaches by Janzing [18]. We discretise $G$ if the method accepts only discrete variables. For IVL regression, we select the regularization parameter $\alpha$ in a variety of ways, including vanilla cross validation (CV), level-based CV (LCV) and confounder correction (CC) as described in Appendix D. Other implementation details are provided in Appendix E, and the code to reproduce our results is publicly released at `https://github.com/uzairakbar/causal-data-augmentation`.

To make CER based evaluation more interpretable for our experiments, we propose the normalization

$$\text{nCER}_{\mathfrak{M}}(h) \coloneqq \frac{\text{CER}_{\mathfrak{M}}(h)}{\text{CER}_{\mathfrak{M}}(h) + \text{CER}_{\mathfrak{M}}(h_0)} \in [0, 1], \qquad h_0(\cdot) \coloneqq \mathbb{E}^{\mathfrak{M};\text{do}(X)}[Y],$$

where $h_0$ represents the null treatment effect, i.e. when $X$ has no causal influence on $Y$, then $\mathbb{E}^{\mathfrak{M};\text{do}(X)}[Y \mid X] = \mathbb{E}^{\mathfrak{M};\text{do}(X)}[Y]$. The normalized CER (nCER) can be considered a generalization of the metrics used by [18, 24, 31] in linear settings and similarly has the interesting property that it is 0 for the ground-truth causal solution $h = f \neq h_0$ but 1 if there is pure confounding for $h \neq f = h_0$. Janzing argues in [24, 31] that using an Euclidean norm instead of the weighted norm in Eq. (8) is more relevant for causal settings, which also motivates our choice when evaluating results of the simulation and optical-device experiments described below. Conceptually, this is equivalent to evaluation based on the causal risk in Eq. (3) under the interventional distribution $X' \sim \hat{\mathcal{N}}(\mathbf{0}_m, \mathbf{I}_m)$.

### 6.1 Simulation experiment

For the finite sample results of the linear SEM $\mathfrak{A}$ from Example 2, by taking $m = 32$, $k = 31$ (dimension of $G$), $\sigma = 0.1$ and fixing $\boldsymbol{\tau}^\top = \mathbf{0}_m$,[9] we sample a new $\mathbf{f}, \boldsymbol{\epsilon}$ and $\mathbf{T} \in \mathbb{R}^{m \times m}$ from a standard normal distribution for each of the 32 experiments for every combination of $\kappa$ and $\gamma$. Each time we construct a $\boldsymbol{\Gamma} \coloneqq \mathbf{V}_0$ with $k$ rows as orthonormal basis of $\text{null}(\mathbf{f})$, such that the SVD of $\mathbf{f}$ is

$$\mathbf{f} = \begin{bmatrix} \mathbf{u} & \mathbf{U}_0 \end{bmatrix} \begin{bmatrix} \lambda & \mathbf{0}_{1 \times (m-1)} \\ \mathbf{0}_{(m-1) \times 1} & \mathbf{0}_{(m-1) \times (m-1)} \end{bmatrix} \begin{bmatrix} \mathbf{v}^\top \\ \mathbf{V}_0^\top \end{bmatrix}.$$

Although this construction of $\boldsymbol{\Gamma}$ relies on direct knowledge of $\mathbf{f}$, which is of course unavailable in practice, we include it here purely for illustrative purposes. We treat access to $\boldsymbol{\Gamma}$ as having prior knowledge about the structural symmetries of $\mathbf{f}$, noting that this information alone is insufficient to recover $\mathbf{f}$.

---

[9]Simulation results are similar under a cyclic setting with a non-trivial $\boldsymbol{\tau}$, and discussed under Appendix E.

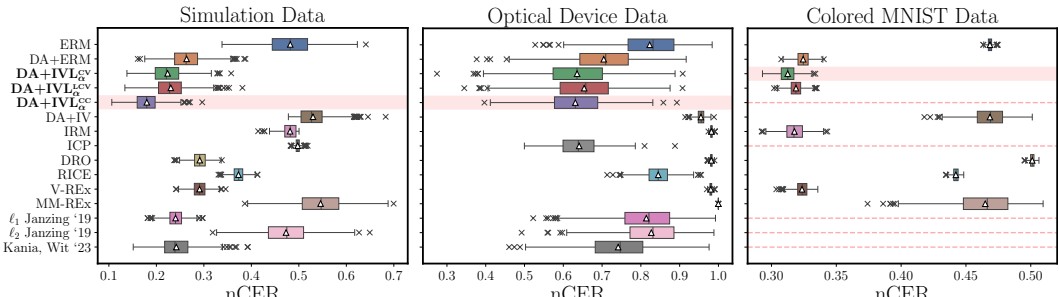

Figure 5: Experiment results; common OOD generalisation benchmarks compared against the ERM, DA+ERM and DA+IV baselines including DA+IVL.

We then generate $n = 2048$ samples of $(X, Y)$ for each experiment. For ERM we use a closed form linear OLS solution. For DA+IV, we make use of linear 2SLS. Finally, DA+IVL$_\alpha$ was implemented using a closed form linear OLS solution between empirical versions (see Proposition 1) of

$$X' := \sqrt{\alpha} X + \left(\sqrt{1+\alpha} - \sqrt{\alpha}\right) \mathbb{E}[X \mid Z], \qquad Y' := \sqrt{\alpha} Y + \left(\sqrt{1+\alpha} - \sqrt{\alpha}\right) \mathbb{E}[Y \mid Z].$$

Our first experimental result in Fig. 4a compares the different estimation methods across varying levels of confounding $\kappa \in [0, 1]$. As expected, ERM performance degrades with increasing confounding. Applying DA alone already brings us closer to the causal solution, while DA+IVL achieves even better performance. DA+IV regression is unstable and generally performs poorly as it is under-determined.

Next, we fix the confounding and DA strengths at $\kappa = \gamma = 1$, and sweep over the regularization parameter $\alpha \in [10^{-5}, 10^5]$ for DA+IVL$_\alpha$. Figure 4b shows that optimal performance is achieved for intermediate values of $\alpha$, confirming that arbitrarily small values of $\alpha$, while beneficial in the theoretical population setting (as suggested by Eq. (27) in the proof of Theorem 2), are suboptimal for finite samples.[10] We also find that both CV and CC strategies effectively select reasonable values of $\alpha$.

Lastly, Fig. 4c examines sensitivity to the DA strength $\gamma \in [10^{-2.5}, 10]$, for fixed confounding strength $\kappa = 1$. As expected, stronger DA results in stronger interventions on $X$, which improves causal effect estimation. However, we also observe diminishing returns; when the variation induced by DA is either too small or too large, DA+IVL$_\alpha$ does not yield significant improvements over the DA+ERM baseline.

For completeness, we also benchmark our approach against other baseline methods on 16 distinct simulation SEMs with 2048 samples each. Aggregated results are presented in Fig. 5 (left most).

## 6.2 Real data experiments

**Optical device dataset.** The dataset from [24] consists of $3 \times 3$ pixel images $X$ displayed on a laptop screen that cause voltage readings $Y$ across a photo-diode. A hidden confounder $C$ controls two LEDs; one affects the webcam capturing $X$, the other affects the photo-diode measuring $Y$. The ground-truth predictor $\mathbf{f}$ is computed by first regressing $Y$ on $(\phi(X), C)$, where $\phi(X)$ are polynomial features of $X$ with degree $d \in \{1, \cdots, 5\}$ that best explains the data (degree 2 in most cases). The component corresponding to $C$ is then removed to recover $\mathbf{f}$. We add Gaussian noise $G \sim \mathcal{N}(\mathbf{0}, \mathbf{\Sigma}_X / 10)$ for DA and fit the methods from Sec. 6.1 on features $\phi(GX)$ for $n = 1000$ samples across 12 datasets. Note that using the same ground-truth polynomial degree for $\phi$ during evaluation is important here so as to avoid introducing statistical bias from model-miss-specification as our analysis squarely focuses on confounding bias. Figure 5 (middle) shows the results, where DA+ERM improves over ERM, and DA+IVL performs even better, outperforming other baselines.

**Colored MNIST.** We evaluate on the colored MNIST dataset [41], where labels are spuriously correlated with image color during training, but this correlation is flipped at test time. We use the same neural architecture and parameters as [41] across all baselines, training with the IV-based objective described in the Appendix C. DA is implemented via small perturbations to hue, brightness, contrast, saturation, and translation, each parameterized by $G \sim \mathcal{B}(2, 2)$. Although these do not

---

[10]We conjecture that this may be due to outcome invariance not holding exactly in practice. A more rigorous investigation is deferred to future work in order to keep the current manuscript more focused.

directly manipulate color, the actual spurious feature, they still help reduce confounding. Results in Fig. 5 (rightmost) show that ERM underperforms, DA+ERM provides substantial gains, and DA+IVL$_\alpha$ performs competitively with the best DG baselines, with DA+IVL$_\alpha^{\text{CV}}$ achieving the best overall performance. Interested readers may also visit Appendix E.3, where we clarify the connection of the colored MNIST model with the cyclic SEM from Eq. (9).

# 7 Limitations

**Necessity and practicality of prior knowledge.**  As discussed in Sec. 4, outcome invariance alone does not suffice to lower confounding bias and practitioners may need domain knowledge to construct DA that targets spurious features as well. Alternatively, one can also take a 'carpet bombing' approach by exhausting all available outcome invariant DA in hope that some may align with spurious features. Nevertheless, under outcome invariance, our methods should perform no worse than standard ERM.

Fundamentally, causal estimation from purely observational data is impossible without untestable assumptions. For instance, the IV (or IVL) assumptions of un-confoundedness and exclusion restriction are inherently untestable and must be justified through domain knowledge. Moreover, the requirement of alignment with spurious features in Theorem 2 is not an artifact of our IVL relaxation—it is a rephrasing of the exclusion principle that underlies identifiability in IV regression. If an IV does not influence $Y$ through the spurious features of $X$, the corresponding causal components of $f$ cannot be identified [25]. IVLs, being relaxations of IVs, inherit these same untestable premises.

Viewed through the lens of IVs/IVLs (Observation 2), our assumptions on DA are arguably more modest than they may initially seem, especially since a symmetry-based DA model has well-established precedent in the literature [3, 28, 53, 57–63]. This correspondence can be summarized as follows:

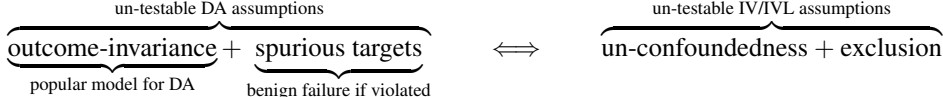

In this light, our framework may in fact be quite practical in domains where valid IVs (or other auxiliary variables) are scarce, but plausible outcome-invariances—i.e., data augmentations—are abundant.

Finally, we recognize the hesitation in committing to strict notions of outcome invariance in practice and leave a more thorough exploration of approximate or even violated invariance to future work.

**Choice of $\alpha$.**  Selecting the IVL regularization parameter $\alpha$ in finite-sample settings is not straightforward. As outlined in Appendix D, we propose several strategies that work well empirically, though some may appear less principled since $\alpha$ is tuned via cross-validation within the same distribution, even though the task concerns OOD generalization. This challenge is not unique to IVL, but rather a broader limitation common to DG methods [64].

# 8 Conclusion

We conclude that our proposed causal framework for data augmentation (DA) enables re-purposing the widely used i.i.d. generalization tool for OOD generalization across treatment interventions. By interpreting outcome-invariant DA as interventions and IV-like variables, our approach reduces confounding bias and consequently improves both causal effect estimation and robust prediction.

# Acknowledgments

To my co-authors for their patience, to Zulfiqar for being my rubber-duck and saving the OpenReview submissions minutes before the deadline, and to all of ML pyos for lightening the chaos with comedy. Thank you.

This work was supported by the NSF (ECCS-2401391, IIS-2403240), and ONR (N000142512173).

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

# Appendix—An Analysis of Causal Effect Estimation using Outcome Invariant Data Augmentation

**Uzair Akbar**
Georgia Tech

**Niki Kilbertus**
TU Munich
Helmholtz AI

**Hao Shen**
TU Munich
Fortiss GmbH

**Krikamol Muandet**
Rational Intelligence
CISPA

**Bo Dai**
Georgia Tech
Google DeepMind

## Contents

## List of Symbols

The notation is largely borrowed from [4], with some overloading where necessary.

| | |
|---|---|
| $\mathbb{R}^{n \times *}$ | $n \times *$ Euclidean space; dimension $*$ conformal with & inferred from context. |
| $x$ | Scalar. |
| $\mathbf{x}$ | Vector. When $\mathbf{x}^\top$ is described as a vector, it means $\mathbf{x}$ is a flat $1 \times *$ matrix. |
| $\mathbf{X}$ | Matrix. |
| $\mathcal{X}$ | Set. |
| $X$ | Random vector. |
| $\mathfrak{M}$ | SEM. |
| $X^{\mathfrak{M}}$ | Random vector $X$ with its SEM $\mathfrak{M}$ specified when unclear from context. |
| $\mathbb{P}_X^{\mathfrak{M}}$ | Distribution of $X$ entailed by $\mathfrak{M}$. Superscript dropped if clear from context. |
| $\mathbf{\Sigma}_X^{\mathfrak{M}}$ | Variance–covariance matrix of $X$ under distribution $\mathbb{P}_X^{\mathfrak{M}}$. |
| $\mathbf{\Sigma}_{X,Y}^{\mathfrak{M}}$ | Cross–covariance matrix of $X$ and $Y$ under distribution $\mathbb{P}_{X,Y}^{\mathfrak{M}}$. |
| $\mathbb{E}^{\mathfrak{M}}[X]$ | Expected value of $X$ under distribution $\mathbb{P}_X^{\mathfrak{M}}$. |
| $\mathrm{do}(X := \mathbf{x})$ | Intervention — $X$ is set to $\mathbf{x}$. |
| $\mathrm{do}(X)$ | Shorthand for $\mathrm{do}(X := X')$ where $X' \sim \mathbb{P}_X^{\mathfrak{M}}$ is i.i.d. to $X$. |
| $\mathfrak{M}; \mathrm{do}(X := \mathbf{x})$ | Intervention SEM. |
| $\mathfrak{M}_{X=\mathbf{x}}$ | SEM with mechanisms of $\mathfrak{M}$, but exogenous noise distribution $\mathbb{P}_{N|X=\mathbf{x}}^{\mathfrak{M}}$. |
| $\mathfrak{M}_{Y=\mathbf{y}}; \mathrm{do}(X := \mathbf{x})$ | Counterfatual SEM—intervention SEM of $\mathfrak{M}_{Y=\mathbf{y}}$. |
| $X \perp\!\!\!\perp Y$ | Random vectors $X, Y$ are statistically independent, i.e. $\mathbb{P}_{Y|X}^{\mathfrak{M}} = \mathbb{P}_Y^{\mathfrak{M}}$. |
| $\mathbf{x} \perp \mathbf{y}$ | $\mathbf{x}, \mathbf{y}$ are perpendicular, i.e. $\mathbf{x}^\top \mathbf{y} = 0$. For random vectors, $X^\top Y = 0$ a.s. |
| $\hat{h}^{\mathfrak{M}}$ | Population/ infinite-sample estimate based on distribution $\mathbb{P}^{\mathfrak{M}}$. |
| $\hat{h}^{\mathcal{D}}$ | Finite-sample estimate based on samples in the dataset $\mathcal{D}$. |

# A    Confounding Bias

**Statistical vs. causal inference.**    The target estimand for the statistical risk in Eq. (2) is the Bayes optimal predictor $\mathbb{E}^{\mathfrak{M}}[Y \mid X = \mathbf{x}]$. And the target estimand for the causal risk in Eq. (3) is the average treatment effect (ATE) $\mathbb{E}^{\mathfrak{M};\mathrm{do}(X:=\mathbf{x})}[Y \mid X = \mathbf{x}] = f(\mathbf{x})$. As such, *statistical inference* is concerned with *predictions* of outcome $Y$, whereas *causal inference* is concerned with *estimating* $f(\mathbf{x})$.

**Statistical vs. confounding bias.**    Both types of inference are subject to bias. *Statistical bias* arises due to miss-specification of the hypothesis class $\mathcal{H}$, whereas confounding bias arises due to how the data are generated. The former is therefore a property of the estimator while the later is a property of the data itself. For an estimator $\hat{h}^{\mathcal{D}}$ with the expected value $\bar{h}(\cdot) = \mathbb{E}^{\mathfrak{M}}_{\mathcal{D}}\left[\hat{h}^{\mathcal{D}}(\cdot)\right]$, we define these as

$$\text{Statistical bias} := \mathbb{E}^{\mathfrak{M}}[Y \mid X = \cdot] - \bar{h}(\cdot),$$
$$\text{Confounding bias} := f(\cdot) - \mathbb{E}^{\mathfrak{M}}[Y \mid X = \cdot].$$

**Bias-variance decomposition of the causal risk.**    Because the treatment $X$ and residual $\xi$ are not correlated under $\mathfrak{M};\mathrm{do}(X)$ in Eq. (1), for any loss function $\ell$ that admits a 'clean' or 'additive' bias-variance decomposition [65], the causal risk in Eq. (3) also admits a bias-variance decomposition. Using squared loss as an example, we have for some hypothesis $\hat{h}^{\mathcal{D}}$,

$$\Rightarrow R^{\mathfrak{M}}_{\mathrm{CR}}\left(\hat{h}^{\mathcal{D}}\right)$$
$$= \mathbb{E}^{\mathfrak{M};\mathrm{do}(X)}\left[\left\| Y - \hat{h}^{\mathcal{D}}(X) \right\|^2\right],$$
$$= \mathbb{E}^{\mathfrak{M};\mathrm{do}(X)}\left[\left\| f(X) + \xi - \hat{h}^{\mathcal{D}}(X) \right\|^2\right], \qquad\qquad \text{(Structural eq. of } Y.)$$
$$= \mathbb{E}^{\mathfrak{M};\mathrm{do}(X)}\left[\|\xi\|^2\right] + \mathbb{E}^{\mathfrak{M};\mathrm{do}(X)}\left[\left\| f(X) - \hat{h}^{\mathcal{D}}(X) \right\|^2\right], \quad \text{(Cross term is 0 as } \xi \perp\!\!\!\perp X^{\mathfrak{M};\mathrm{do}(X)}.)$$
$$= \underbrace{\mathbb{E}^{\mathfrak{M};\mathrm{do}(X)}\left[\|\xi\|^2\right]}_{\text{irreducible noise}} + \underbrace{\mathbb{E}^{\mathfrak{M}}\left[\left\| f(X) - \hat{h}^{\mathcal{D}}(X) \right\|^2\right]}_{\text{estimation error, } \mathrm{CER}_{\mathfrak{M}}\left(\hat{h}^{\mathcal{D}}\right)=}. \quad (\mathbb{P}^{\mathfrak{M}}_X, \mathbb{P}^{\mathfrak{M};\mathrm{do}(X)}_X \text{ identical by construction.})$$

We can show by following standard procedure that

$$\mathbb{E}^{\mathfrak{M}}_{\mathcal{D}}\left[\mathrm{CER}_{\mathfrak{M}}\left(\hat{h}^{\mathcal{D}}\right)\right] = \underbrace{\mathbb{E}^{\mathfrak{M}}_X\left[\left\| f(X) - \bar{h}(X) \right\|^2\right]}_{\text{bias}^2} + \underbrace{\mathbb{E}^{\mathfrak{M}}_{\mathcal{D}}\left[\mathbb{E}^{\mathfrak{M}}_X\left[\left\| \bar{h}(X) - \hat{h}^{\mathcal{D}}(X) \right\|^2\right]\right]}_{\text{variance}}.$$

Since for any population estimate $\hat{h}^{\mathfrak{M}}(X) = \bar{h}(X)$, the CER equals the average (squared) bias in estimation

$$\mathrm{CER}_{\mathfrak{M}}\left(\hat{h}^{\mathfrak{M}}\right) = \mathbb{E}^{\mathfrak{M}}_X\left[\left\| f(X) - \hat{h}^{\mathfrak{M}}(X) \right\|^2\right] = \mathbb{E}^{\mathfrak{M}}_X\left[\left\| f(X) - \bar{h}(X) \right\|^2\right].$$

For a rich enough hypothesis class, the ERM estimate coincides with the Bayes optimal predictor $\hat{h}^{\mathfrak{M}}_{\mathrm{ERM}}(\cdot) = \mathbb{E}^{\mathfrak{M}}[Y \mid X = \cdot]$ and the CER exactly equals the (average squared) confounding bias as we define it above. For a general estimate $\hat{h}^{\mathcal{D}}$, however, the CER also contains statistical bias. Nevertheless, our claims of "better causal estimation via reducing confounding bias" rest on the fact that we are essentially manipulating the data via DA and/or using treatment randomization sources in the form of IVLs. And recall that confounding bias is a property of the data.

# B    Simultaneity as Cyclic Structures in Equilibrium

**Linear cyclic assignments**

SEMs with cyclic structures have been well studied both in the linear case [66–68], as well as the non-linear case [69, 70]. Here we briefly provide a causal interpretation to linear simultaneous equations as SEMs with cyclic assignments.

Consider a square matrix $\mathbf{M} \in \mathbb{R}^{d \times d}$ and the SEM

$$W = \mathbf{M}W + N \ , \tag{10}$$

where random noise vector $N$ is exogenous and $\mathbf{M}$ allows for a cyclic structure. We enforce $(\mathbf{I}_d - \mathbf{M})$ to be invertible so that the above equation has a unique solution $W$ for any given $N$. Re-writing the *structural form* in Eq. (10) into a *reduced form*, the distribution over $W$ is defined by

$$W = (\mathbf{I}_d - \mathbf{M})^{-1} N \ . \tag{11}$$

One way we can present a causal interpretation of the above solution is to view it as a stationary point to the following sequence of random vectors $W_t$

$$W_t = \mathbf{M}W_{t-1} + N \ ,$$

which converges if $\mathbf{M}$ has a spectral norm strictly smaller than one so that $\mathbf{M}^t \to 0$ as $t \to \infty$. The structural form Eq. (10) essentially describes the iterative application of this operation. And in the limit the distribution of $\lim_{t \to \infty} W^t$ will be the same as the reduced form Eq. (11). Although equivalent, reduced form of a cyclic SEM (if one exists) obscures the causal relations in the data generation process.

Furthermore, we restrict our models to not have any "self-cycles" (an edge from a vertex to itself). So, e.g., the matrix $\mathbf{M}$ in Eq. (10) has all zero diagonal entries. This not only simplifies our analysis by providing a simple and intuitive interpretation for our definition of DA in Sec. 2.3, but it also ensures that non-linear SEMs entail unique, well-defined distributions under mild assumptions [70, 67].

Similarly we can write the example SEM $\mathfrak{M}$ from Example 1 in this (block matrix) form as

$$\underbrace{\begin{bmatrix} X \\ Y \end{bmatrix}}_{W} = \underbrace{\begin{bmatrix} \mathbf{0}_{m \times m} & \boldsymbol{\tau}^\top \\ \mathbf{f}^\top & \mathbf{0}_{1 \times 1} \end{bmatrix}}_{\mathbf{M}} \underbrace{\begin{bmatrix} X \\ Y \end{bmatrix}}_{W} + \underbrace{\begin{bmatrix} \boldsymbol{\Gamma}^\top \\ \mathbf{0}_{1 \times k} \end{bmatrix} Z + \begin{bmatrix} \mathbf{T}^\top \\ \boldsymbol{\epsilon}^\top \end{bmatrix} C + \sigma \cdot \begin{bmatrix} N_X \\ N_Y \end{bmatrix}}_{N},$$

For this simple case, $\left( \mathbf{I}_{(m+1)} - \mathbf{M} \right)$ is always invertible so long as $\mathbf{f}^\top \boldsymbol{\tau}^\top \neq 1$ from Lemma 3. Or we can also restrict $\left| \mathbf{f}^\top \boldsymbol{\tau}^\top \right| < 1$ to ensure that the spectral norm of $\mathbf{M}$ is strictly smaller than 1. We sample from this SEM by first sampling all of the exogenous variables $Z, C, N_X, N_Y$ and then solving the above system for each sample of $X, Y$ via the reduced form in Lemma 3.

**A motivating example**

Cyclic SEMs were first discussed in the econometrics literature [71] to model various observational phenomena, and often solved via 2SLS based IV regression [22] since it is computationally less costly compared to solving the entire system [27]. A classic example from economics [72, 73] is that of a *supply and demand model* $\mathfrak{M}$ where the relation of price $P$ of a good with quantity $Q$ of demand can be thought of as a cyclic feed-back loop where producers adjust their price in response to demand of the good and consumers change their demand in response to price of a good. In contrast, a change in consumer tastes or preferences would be an exogenous change on the demand curve and can therefore be used as an IV $Z$.

$$\text{consumer demand:} \quad Q = \tau \cdot P + \gamma \cdot Z + N_Q \ ,$$
$$\text{producer price:} \quad P = f \cdot Q + N_P \ .$$

Where scalars $f, \tau$ are such that $|f \cdot \tau| < 1$ so that the system converges to an equilibrium. We say that the measurements made for $P$ and $Q$ are at the equilibrium state of the market[11] with zero mean measurement noise $N_P, N_Q$ respectively.

---

[11]In fact, such a feed-back model of supply and demand was initially developed to understand the irregular fluctuations of prices/quantities that are observed in some markets when not at equilibrium [72].

**Mitigating simultaneity bias for causal effect estimation.** If we now want to *estimate* the effect of demand on price $f$, standard regression will produce a biased estimate $\hat{f}_{\text{ERM}}^{\mathfrak{M}} = f + \frac{\text{Cov}(Q, N_P)}{\text{Var}(Q)}$ because of the simultaneity causing $Q$ and $N_P$ to be correlated (to see this, substitute model of $P$ into the model of $Q$). We can now use IV regression to get an unbiased estimate of the effect of demand on price in the market as $\hat{f}_{\text{IV}}^{\mathfrak{M}} = f$.

**Mitigating spurious correlations for robust prediction.** Similarly, if the producer wants to *predict* the effect on demand if price is changed (i.e. intervened on), naive ERM will not be a good choice because it will also capture the spurious correlation from $Q \to P$. We therefore use three-stage-least-squares (3SLS) [74, 27] (or similar methods) to estimate the ATE $\hat{\tau}_{\text{3SLS}}^{\mathfrak{M}} = \mathbb{E}^{\mathfrak{M};\text{do}(P:=\cdot)}[Q \mid P = \cdot]$ where we use the first two stages to estimate $\hat{f}_{\text{IV}}^{\mathfrak{M}}$, followed by ERM to regress from the residuals $\hat{N}_P := P - \hat{f}_{\text{IV}}^{\mathfrak{M}} \cdot Q$ to $Q$ in the third stage.

### Implications for independence of causal mechanisms

Here we clarify how the equilibrium assumption/interpretation of cyclic SEMs is not at odds with the classic independent causal mechanism (ICM) principle [4]. Note that our SEM formulation in Eq. (1) is a direct instantiation of the ICM principle as described by Peters et al. [4]. The two equations represent the autonomous mechanisms, and their independence is captured by the mutual independence of the exogenous noise terms $N_X, N_Y$. The simultaneity in our model is not a violation of ICM, but rather the equilibrium state resulting from the interaction of these two independent mechanisms. Assuming the existence of this equilibrium is a statement about the scope of systems under analysis, and not about the nature of the mechanisms themselves. Indeed, surgically changing $\tau$ to some $\tau'$, for example, does not in itself alter $f$ and vice versa. And precisely because of the ICM, this may or may not make the system unstable depending on the nature of $\tau'$. Nevertheless, in our setting, Proposition 2 (Appendix F.2) shows that soft interventions induced by outcome-invariant DA are *always* stable.

## C  IV Regression Supplement

**Two-stage estimators.** Minimizing the risk in Eq. (5) is known as two-stage IV regression. Another two-stage IV regression approach that we use in our theoretical results is to minimize the risk [8, 15]

$$R_{\text{IV}_{\text{LB}}}^{\mathfrak{M}}(h) := \mathbb{E}^{\mathfrak{M}}\left[\left\|\mathbb{E}^{\mathfrak{M}}[Y \mid Z] - \mathbb{E}^{\mathfrak{M}}[h(X) \mid Z]\right\|^2\right].$$

This can be shown to lower-bound (hence the subscript LB) the risk in Eq. (5) under squared loss [8].

$$\Rightarrow R_{\text{IV}}^{\mathfrak{M}}(h) = \mathbb{E}\left[\|Y - \mathbb{E}[h(X) \mid Z]\|^2\right],$$

$$= \mathbb{E}\left[\|(Y - \mathbb{E}[Y \mid Z]) + (\mathbb{E}[Y \mid Z] - \mathbb{E}[h(X) \mid Z])\|^2\right], \quad \text{(Adding and subtracting } \mathbb{E}[Y \mid Z].)$$

$$= \mathbb{E}\left[\|Y - \mathbb{E}[Y \mid Z]\|^2\right] + \mathbb{E}\left[\|\mathbb{E}[Y \mid Z] - \mathbb{E}[h(X) \mid Z]\|^2\right] \quad \text{(Expand squared norm.)}$$

$$\qquad + 2\mathbb{E}\left[(Y - \mathbb{E}[Y \mid Z])^\top (\mathbb{E}[Y \mid Z] - \mathbb{E}[h(X) \mid Z])\right],$$

$$= \mathbb{E}\left[\|Y - \mathbb{E}[Y \mid Z]\|^2\right] + \mathbb{E}\left[\|\mathbb{E}[Y \mid Z] - \mathbb{E}[h(X) \mid Z]\|^2\right], \tag{12}$$

$$= \mathbb{E}\left[\|\mathbb{E}[Y \mid Z] - \mathbb{E}[h(X) \mid Z]\|^2\right] + \mathbb{E}\left[\mathbb{E}\left[(Y - \mathbb{E}[Y \mid Z])^2 \mid Z\right]\right], \quad \text{(Tower rule, scalar } Y.)$$

$$= \mathbb{E}\left[\|\mathbb{E}[Y \mid Z] - \mathbb{E}[h(X) \mid Z]\|^2\right] + \mathbb{E}[\text{Var}(Y \mid Z)] = R_{\text{IV}_{\text{LB}}}^{\mathfrak{M}}(h) + \mathbb{E}[\text{Var}(Y \mid Z)], \tag{13}$$

where Eq. (13) follows from the definition of conditional variance and we get Eq. (12) by setting the cross term to zero since

$$\Rightarrow \mathbb{E}\left[(Y - \mathbb{E}[Y \mid Z])^\top (\mathbb{E}[Y \mid Z] - \mathbb{E}[h(X) \mid Z])\right]$$

$$= \mathbb{E}\left[\mathbb{E}\left[(Y - \mathbb{E}[Y \mid Z])^\top (\mathbb{E}[Y \mid Z] - \mathbb{E}[h(X) \mid Z]) \mid Z\right]\right], \quad \text{(Tower rule.)}$$

$$= \mathbb{E}\left[\mathbb{E}\left[(Y - \mathbb{E}[Y \mid Z])^\top \mid Z\right](\mathbb{E}[Y \mid Z] - \mathbb{E}[h(X) \mid Z])\right], \tag{14}$$

$$= \mathbb{E}\left[(\mathbb{E}[Y \mid Z] - \mathbb{E}[Y \mid Z])^\top (\mathbb{E}[Y \mid Z] - \mathbb{E}[h(X) \mid Z])\right],$$

$$= \mathbb{E}\left[\mathbf{0}^\top (\mathbb{E}[Y \mid Z] - \mathbb{E}[h(X) \mid Z])\right] = 0,$$

where Eq. (14) follows from the "taking out what is known" rule, i.e.,

$$\mathbb{E}[g(B)A \mid B] = g(B)\mathbb{E}[A \mid B]. \tag{15}$$

**Generalized method of moments.** The IV regression in our colored-MNIST experiment uses the popular *generalized methods of moments (GMM)* [75–77], or equivalently the *conditional moment restriction (CMR)* [8] framework which tries to directly solve for the fact that in Eq. (1) with scalar $Y$

$$\mathbb{E}^{\mathfrak{M}}[\xi \mid Z] = \mathbb{E}^{\mathfrak{M}}[Y - f(X) \mid Z] = 0,$$

which holds as a direct consequence of un-confoundedness of $Z$. For any $q : \mathcal{Z} \to \mathbb{R}$, it then follows

$$\mathbb{E}^{\mathfrak{M}}\left[(Y - f(X)) \cdot q(Z)\right] = 0 .$$

The GMM-IV estimate of $f$ therefore tries to enforce this condition [75–77] by minimizing the risk

$$R_{\text{IV}_{\text{GMM}}}^{\mathfrak{M}}(h) := \sum_{i=1}^{\mu} \mathbb{E}^{\mathfrak{M}}\left[(Y - h(X)) \cdot q_i(Z)\right]^2 = \left\|\mathbb{E}^{\mathfrak{M}}[(Y - h(X)) \cdot \mathbf{q}(Z)]\right\|^2,$$

where $\mathbf{q}(\cdot) \in \mathbb{R}^\mu$ represents a vector form of the set of $\mu$ arbitrary real-valued functions $q_i$. A more general form of the above GMM based IV risk is to weight the norm by some SPD $\mathbf{W}$ [78, 75, 76]

$$R_{\text{IV}_{\text{GMM-}\mathbf{W}}}^{\mathfrak{M}}(h) := \left\|\mathbb{E}^{\mathfrak{M}}[(Y - h(X)) \cdot \mathbf{q}(Z)]\right\|_{\mathbf{W}}^2,$$

which gives the most statistically efficient estimator, minimizing the asymptotic variance, for $\mathbf{W} = \mathbf{\Sigma}_Z^{-1}$ [78, 75, 76]. We use the same for our colored-MNIST experiments, together with the identity function $\mathbf{q}(Z) = Z$. This gives us the final loss of the form

$$R_{\text{IV}_{\text{GMM-}\mathbf{\Sigma}_Z^{-1}}}^{\mathfrak{M}}(h) = \left\|\mathbb{E}^{\mathfrak{M}}[Z \cdot (Y - h(X))]\right\|_{\mathbf{\Sigma}_Z^{-1}}^2.$$

And the empirical version of which can be written as follows

$$R^{\mathcal{D}}_{\mathrm{IV}_{\mathrm{GMM}\text{-}\boldsymbol{\Sigma}_Z^{-1}}}(h) := \left(\hat{\mathbf{y}} - \mathbf{h}\left(\hat{\mathbf{X}}\right)\right)^{\top} \hat{\mathbf{Z}}\hat{\mathbf{Z}}^{\dagger}\left(\hat{\mathbf{y}} - \mathbf{h}\left(\hat{\mathbf{X}}\right)\right),\tag{16}$$

where for dataset samples $(\mathbf{x}_i, y_i, \mathbf{z}_i) \in \mathcal{D}$, we construct the vector $\hat{\mathbf{y}} := [y_0, \cdots, y_n]^{\top}$, matrices $\hat{\mathbf{X}} := [\mathbf{x}_0^{\top}, \cdots, \mathbf{x}_n^{\top}]^{\top}$, $\hat{\mathbf{Z}} := [\mathbf{z}_0 \quad \cdots \quad \mathbf{z}_n]^{\top}$ with pseudo-inverse $\hat{\mathbf{Z}}^{\dagger}$ and define $\mathbf{h}\left(\hat{\mathbf{X}}\right) := [h(\mathbf{x}_0), \cdots, h(\mathbf{x}_n)]^{\top}$.

# D  IVL Regression Supplement

**Closed form solution in the linear case.**    The following result gives us a way to compute a closed-form solution to the $\text{IVL}_\alpha$ regression problem in the linear Gaussian case. An empirical version of this is used for our linear experiments.

**Proposition 1** ($\text{IVL}_\alpha$ closed form solution). *For SEM $\mathfrak{M}$ in Example 1, $\hat{\text{h}}_{IVL_\alpha}^{\mathfrak{M}}$ is the closed form linear OLS solution between*

$$X' := aX + b\mathbb{E}[X \mid Z], \qquad\qquad Y' := aY + b\mathbb{E}[Y \mid Z],$$

*where*

$$a := \sqrt{\alpha}, \qquad\qquad b := \sqrt{1+\alpha} - \sqrt{\alpha}.$$

*Proof.* See Appendix F.1 for the proof. $\qquad\qquad\qquad\qquad\qquad\qquad\qquad\qquad\qquad$ $\square$

For the empirical version of Proposition 1 we fit a closed-form OLS regressor between

$$X' := \sqrt{\alpha}X + \left(\sqrt{1+\alpha} - \sqrt{\alpha}\right)\hat{\mathbf{Z}}\hat{\mathbf{Z}}^\dagger X, \qquad Y' := \sqrt{\alpha}Y + \left(\sqrt{1+\alpha} - \sqrt{\alpha}\right)\hat{\mathbf{Z}}\hat{\mathbf{Z}}^\dagger Y,$$

where $\hat{\mathbf{Z}}, \hat{\mathbf{Z}}^\dagger$ are as defined in Eq. (16).

**Choice of regularization parameter.**    We try the following approaches to select the parameter $\alpha$.

*Cross validation (CV)*, or any variation thereof. We specifically use the following two in our experiments; (i) vanilla CV with $20\%$ samples held-out for validation (ii) *level cross validation (LCV)* for when $Z$ is discrete, where hold-out data corresponding to $20\%$ of the levels of $Z$ for validation.

*Confounder correction (CC)*, where in a linear setting we follow an approach similar to [18] by estimating the length of the true solution $f$ from the observational data $\mathcal{D}$. We then chose $\alpha$ such that the length of $\hat{h}_{\text{DA}+\text{IVL}_\alpha}^{\mathcal{D}}$ is closest to the estimated length of the ground truth solution.

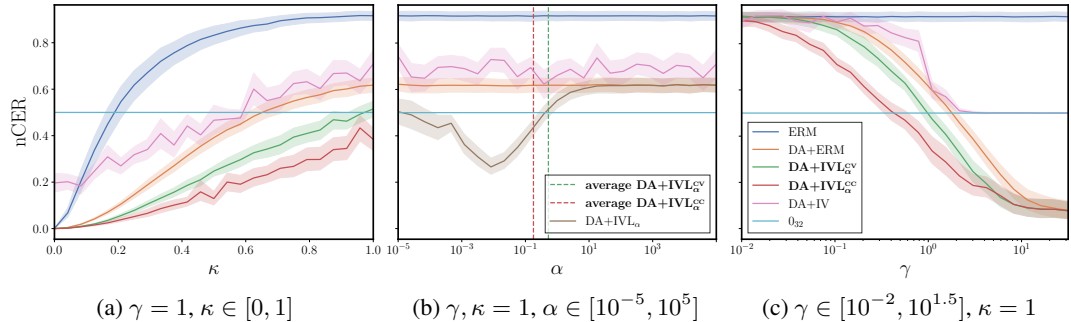

(a) $\gamma = 1, \kappa \in [0, 1]$      (b) $\gamma, \kappa = 1, \alpha \in [10^{-5}, 10^5]$      (c) $\gamma \in [10^{-2}, 10^{1.5}], \kappa = 1$

Figure 6: Simulation of the linear Gaussian SEM of Example 2 with the same setting as Fig. 4, but $\tau^\top, \mathbf{f}$ sampled uniformly over a unit sphere, representing a cyclic structure. Each data-point represents the average nCER over 25 trials with a 95% CI.

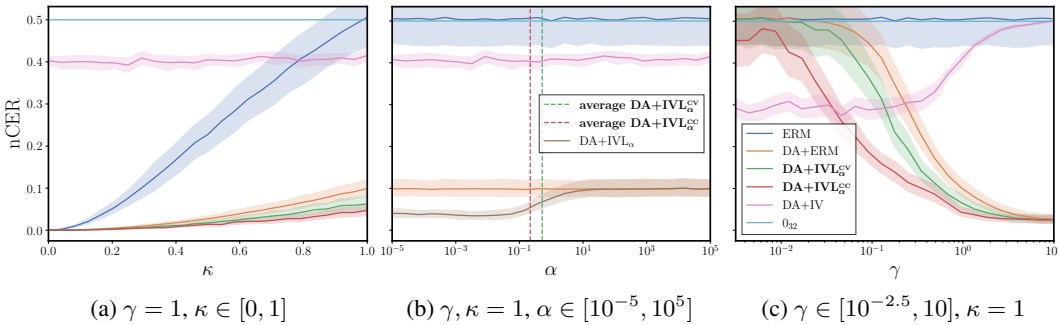

(a) $\gamma = 1, \kappa \in [0, 1]$      (b) $\gamma, \kappa = 1, \alpha \in [10^{-5}, 10^5]$      (c) $\gamma \in [10^{-2.5}, 10], \kappa = 1$

Figure 7: Same experiment as Fig. 4, but with $\mathbf{\Gamma}$ constructed by randomly selecting each basis of $\text{null}(\mathbf{f}^\top)$ with a probability of $2/3$, simulating the effect of knowing only *some* symmetries of $\mathbf{f}$. Each data-point represents the average nCER over 25 trials with a 95% CI.

## E    Experiment Supplement

For the methods that use *stochastic gradient descent (SGD)*, we use a learning rate of $0.01$, batch size of $256$ for $16$ epochs. For baselines that require a discrete domains/environments, we uniformly discretise each dimension of $G$ into 2 bins. Higher discretisation bins renders most baselines ineffective since each domain/environment rarely has more than 1 sample. To keep the comparison fair, however, we also discretize $G$ for $\text{IVL}_\alpha$ regression when using LCV. For the colored MNIST experiment, all CV implementations including baselines use 5-folds for a random search over an exponentially distributed regularization parameter with rate parameter of 1. Same is the case for simulation and optical device experiments, except that DA+IVL methods use a log-uniform distributed regularization parameter over $[10^{-4}, 1]$. Since RICE [56] grows the dataset size by augmenting each sample $T$ times, we provide it a $1/T$ sub-sample of the original data for fair comparison. Similarly, the causal regularization method by Kania and Wit [12] expects two datasets, a perturbed and an un-perturbed one, which we substitute with 1/2 augmented data and 1/2 original data respectively.

### E.1    Simulation experiment

For the parameter sweep experiments of Fig. 4, we generate a treatment of dimension $m = 32$, but for the OOD baseline comparison experiment in Fig. 5 we use $m = 16$. Furthermore, for the OOD baseline comparison experiment in Fig. 5, we randomly pick each basis of $\text{null}(\mathbf{f})$ with a probability $1/3$ to construct $\mathbf{\Gamma}$ (i.e., we know only some, but not all symmetries of $\mathbf{f}$).

We also provide additional linear simulation experiment results in Figs. 6 and 7—the former simulates a cyclic structure with a non-zero $\tau$, and the later simulates a case where only some, but not all symmetries of $\mathbf{f}$ are known. The results of both are consistent with our original experiment in Fig. 4.

Table 2: nCER $\pm$ one standard error (SE) across the 12 optical-device datasets for various choices of DA. **Bold** and *italic* denote the lowest and second-lowest average nCER, respectively. Superscripts $*$ and $\dagger$ indicate a significant improvement over ERM or *both* ERM *and* DA+ERM, respectively, beyond a margin of SE. Lastly, — indicates that the method was too expensive for the value to be computed.

| Method | rotate > hflip > vflip | random-permutation | gaussian-noise | all |
|---|---|---|---|---|
| ERM | $0.827 \pm 0.079$ | $0.827 \pm 0.079$ | $0.827 \pm 0.079$ | $0.823 \pm 0.083$ |
| DA+ERM | $0.617 \pm 0.085^*$ | $\mathbf{0.513 \pm 0.082^*}$ | $0.707 \pm 0.090^*$ | $0.513 \pm 0.075^*$ |
| **DA+IVL$_\alpha^{cv}$** | $0.623 \pm 0.087^*$ | $0.540 \pm 0.085^*$ | $0.641 \pm 0.092^*$ | $0.533 \pm 0.083^*$ |
| **DA+IVL$_\alpha^{Lcv}$** | $0.619 \pm 0.087^*$ | $0.534 \pm 0.082^*$ | $0.662 \pm 0.091^*$ | $0.574 \pm 0.087^*$ |
| **DA+IVL$_\alpha^{cc}$** | $0.623 \pm 0.085^*$ | $0.527 \pm 0.082^*$ | $\mathbf{0.639 \pm 0.076^*}$ | $\mathbf{0.509 \pm 0.078^*}$ |
| DA+IV | $0.689 \pm 0.065^*$ | $0.973 \pm 0.011$ | $0.955 \pm 0.011$ | $0.640 \pm 0.083^*$ |
| IRM | $0.972 \pm 0.010$ | $0.960 \pm 0.015$ | $0.970 \pm 0.009$ | $0.953 \pm 0.018$ |
| ICP | $\mathbf{0.544 \pm 0.019^\dagger}$ | $0.527 \pm 0.012^*$ | $0.646 \pm 0.054^\dagger$ | — |
| DRO | $0.975 \pm 0.005$ | $0.959 \pm 0.012$ | $0.981 \pm 0.003$ | $0.952 \pm 0.014$ |
| RICE | $0.966 \pm 0.014$ | $0.960 \pm 0.012$ | $0.974 \pm 0.005$ | $0.959 \pm 0.016$ |
| V-REx | $0.962 \pm 0.024$ | $0.957 \pm 0.013$ | $0.979 \pm 0.005$ | $0.925 \pm 0.037$ |
| MM-REx | $0.978 \pm 0.013$ | $1.000 \pm 0.000$ | $1.000 \pm 0.000$ | $1.000 \pm 0.000$ |
| $\ell_1$ Janzing '19 | $0.821 \pm 0.081$ | $0.821 \pm 0.081$ | $0.821 \pm 0.081$ | $0.817 \pm 0.077$ |
| $\ell_2$ Janzing '19 | $0.823 \pm 0.076$ | $0.823 \pm 0.076$ | $0.823 \pm 0.076$ | $0.828 \pm 0.079$ |
| Kania, Wit '23 | $0.652 \pm 0.084^*$ | $0.559 \pm 0.084^*$ | $0.727 \pm 0.088^*$ | $0.543 \pm 0.080^*$ |

### E.2  Optical device experiment

In the simulation and optical device experiments, we fit a linear function $h(.) \coloneqq \mathbf{h} \in \mathbb{R}^m$ for a squared loss in all of our risk metrics. For IVL$_\alpha$ regression, we use the closed-form OLS solution from Appendix D. We also use a closed-form solution for ERM, DA+ERM and DA+IV (2SLS) baselines. The rest of the baselines (other than ICP) use SGD.

In Tab. 2, we report further experiments on the optical device dataset with various DA choices. The findings continue to confirm our main hypothesis: DA+IVL dominates DA+ERM, which itself dominates ERM. We never observe an opposite trend with statistical significance.

### E.3  Colored-MNIST experiment

In the colored MNIST experiment, we use the same 3-layer neural network (NN) architecture for $h$ across all methods comprising of a fully-connected input layer of input dimension $m$, hidden layer of input/output dimension 256 and output classification layer with a Sigmoid function. Each layer is separated by an intermediary *rectified linear unit* activation function. For the IV risk, we use the empirical version of the GMM based risk from Eq. (16).

**Colored-MNIST as a cyclic SEM—From invariant prediction to estimating causal effects**

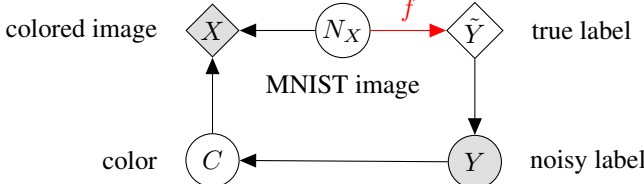

Figure 8: The data generation DAG for colored-MNIST as discussed by the original authors [41]. They aim to learn a predictor $h : \mathcal{X} \to \mathcal{Y}$ such that it is invariant to changes in $\mathbb{P}_{X|Y}$. We argue that this DAG view of colored-MNIST does not make it obvious how the true labeling function $f(\mathbf{x})$ is related to the ATE $\mathbb{E}^{\mathfrak{M}; \mathrm{do}(X \coloneqq \mathbf{x})}[Y \mid X = \mathbf{x}]$, which we believe is because it is virtually equivalent to the reduced form of our structural form presented in Fig. 9.

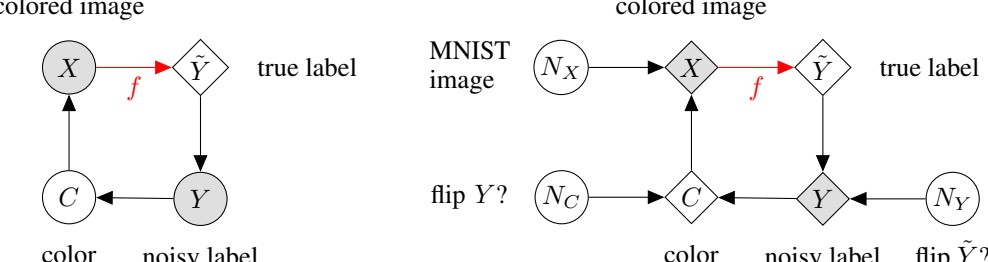

(a) Graph for generating colored-MNIST data.    (b) Augmented graph—exogenous variables explicitly shown.

Figure 9: A cyclic SEM perspective of the colored-MNIST data—an MNIST image $N_X$ is assigned color $C$ to produce a colored-MNIST image $X$. This is then passed through the ground-truth labeling function $f$ to produce the true label $\tilde{Y}$. We flip this with probability 0.25 to produce the observed label $Y$, which in turn is flipped with probability $e$ (at train time $e \in \{0.1, 0.2\}$ and $e = 0.9$ at test time) to produce the color $C$. These assignments are iteratively applied for any joint sample of the exogenous variables $N_X, N_Y, N_C$ starting at arbitrary values of endogenous variables until convergence to the unique stationary point $X, Y, C$ (and $\tilde{Y}$).

In this section we give a cyclic SEM perspective of the colored-MNIST experiment from [41]. The task is binary classification of colored images $X$ from the MNIST dataset into low digits ($y = 0$ for digits from 0 to 4) and high digits ($y = 1$ for digits from 5 to 9). The difficulty of the task arises from there being a higher spurious correlation between the color $C$ of the images ($c = 0$ for blue and $c = 1$ for green) and (noisy) labels $Y$ as compared to the correlation between the digits in the image and the label.

Consider the following cyclic SEM in Fig. 9.

$$\mathbf{n}_X \sim \mathbb{P}_{N_X}, n_Y \sim \mathbb{B}(0.25), n_c \sim \mathbb{B}(e) \quad \text{sample all exogenous variables}$$
$$X = \texttt{colour}(C, \mathbf{n}_X) \quad \text{apply color } C \text{ to the image}$$
$$\tilde{Y} = f(X) \quad \text{generate ground-truth label with true labeling function}$$
$$Y = \texttt{xor}\left(\tilde{Y}, n_Y\right) \quad \text{flip the label with probability 0.25}$$
$$C = \texttt{xor}(Y, n_C) \quad \text{generate color by flipping } Y \text{ with probability } e,$$

where we first randomly sample an un-colored MNIST image $\mathbf{n}_X$, and some Bernoulli distributed label noise $n_Y \sim \mathbb{B}(0.25)$ and color noise $n_C \sim \mathbb{B}(e)$ which is different for each environment $e \in \{0.1, 0.2\}$. Then for some initial arbitrary values $\mathbf{x}_0, \tilde{y}_0, y_0$ and $c_0$ respectively for the observed colored image $X$, the ground-truth label $\tilde{Y}$, the observed noisy label $Y$ and the image color $C$, we iteratively apply the following assignments from the SEM

$$\mathbf{x}_t = \texttt{colour}(c_{t-1}, \mathbf{n}_X) \quad \text{apply color } C \text{ to the image}$$
$$\tilde{y}_t = f(\mathbf{x}_{t-1}) \quad \text{generate ground-truth label with true labeling function}$$
$$y_t = \texttt{xor}(\tilde{y}_{t-1}, n_Y) \quad \text{flip the label with probability 0.25}$$
$$c_t = \texttt{xor}(y_{t-1}, n_C) \quad \text{generate color by flipping } Y \text{ with probability } e,$$

until they converge while keeping all sampled exogenous variables $\mathbf{n}_X, n_Y, n_C$ fixed. It is straightforward to show that this SEM will converge after a maximum of $t = 5$ iterations[12] due to the invariance of $f$ to the color of the image $C$. Furthermore, this stationary-point will be uniquely determined by our exogenous samples $\mathbf{n}_X, n_Y, n_C$. And this is how we generate one sample $(\mathbf{x}, y)$ for our colored-MNIST experiment. We repeat this process to generate a sample $(\mathbf{x}, y)$ for each of $n$ samples $\mathbf{n}_X, n_Y, n_C$.

Note that the ground-truth labeling function $f$ can only correctly predict the labels 75% of the time. At test time we flip the correlation between the label $Y$ and the image color $C$ by setting $e = 0.9$. Also, the above cyclic SEM for colored-MNIST produces the same distribution for $(X, Y)$ as [41].

---

[12]Following the mechanisms $c_0 \to \mathbf{x}_1 \to \tilde{y}_2 \to y_3 \to c_4 \to \mathbf{x}_5$, we see that $(\mathbf{x}_4, y_4, c_4) = (\mathbf{x}_5, y_5, c_5)$ (same for $\tilde{y}_4 = \tilde{y}_5$).

The above cyclic SEM perspective of colored-MNIST is interesting because it makes it clear that colored-MNIST is essentially a causal effect estimation task. Specifically, we can estimate the true labeling function $f$ by estimating the ATE $\mathbb{E}^{\mathfrak{M};\mathrm{do}(X:=\mathbf{x})}[Y \mid X = \mathbf{x}]$ since

$$
\begin{aligned}
\mathbb{E}^{\mathfrak{M};\mathrm{do}(X:=\mathbf{x})}[Y \mid X = \mathbf{x}] &= \mathbb{E}^{\mathfrak{M};\mathrm{do}(X:=\mathbf{x})}[\mathtt{xor}(f(X), N_Y) \mid X = \mathbf{x}], \\
&= \mathbb{E}^{\mathfrak{M}}[\mathtt{xor}(f(\mathbf{x}), N_Y)], && (N_Y \perp\!\!\!\perp X^{\mathfrak{M};\mathrm{do}(X:=\mathbf{x})}.) \\
&= \mathbb{E}^{\mathfrak{M}}[f(\mathbf{x}) + N_Y - 2f(\mathbf{x})N_Y], && (\text{Definition of } \mathtt{xor}.) \\
&= f(\mathbf{x}) + \mathbb{E}^{\mathfrak{M}}[N_Y] - 2f(\mathbf{x})\mathbb{E}^{\mathfrak{M}}[N_Y], \\
&= \left(1 - 2\mathbb{E}^{\mathfrak{M}}[N_Y]\right)f(\mathbf{x}) + \mathbb{E}^{\mathfrak{M}}[N_Y], \\
&= 0.5f(\mathbf{x}) + 0.25 . && (N_Y \sim B(0.25).)
\end{aligned}
$$

Because this is a binary classification task, we have

$$
\mathtt{round}\left(\mathbb{E}^{\mathfrak{M};\mathrm{do}(X:=\mathbf{x})}[Y \mid X = \mathbf{x}]\right) = f(\mathbf{x}).
$$

This is in contrast to the original DAG perspective of colored-MNIST shown in Fig. 8, where the connection to the estimation of the causal mechanism $f$ is not immediately obvious. We argue that this is because the DAG in Fig. 8 is virtually equivalent to the reduced form of our structural form presented in Fig. 9.

# F Proofs

## F.1 Proof of Proposition 1—IVL regression closed form solution in the linear case

**Proposition 1** (IVL$_\alpha$ closed form solution). *For SEM $\mathfrak{M}$ in Example 1, $\hat{\mathbf{h}}_{IVL_\alpha}^{\mathfrak{M}}$ is the closed form linear OLS solution between*

$$X' := aX + b\mathbb{E}[X \mid Z], \qquad\qquad Y' := aY + b\mathbb{E}[Y \mid Z],$$

*where*

$$a := \sqrt{\alpha}, \qquad\qquad b := \sqrt{1+\alpha} - \sqrt{\alpha}.$$

*Proof.* The OLS solution for $(X', Y')$ minimizes the following ERM risk

$$\Rightarrow \mathbb{E}\left[\left\|Y' - \mathbf{h}^\top X'\right\|^2\right]$$

$$= \mathbb{E}\left[\left\|aY + b\mathbb{E}[Y \mid Z] - \mathbf{h}^\top(aX + b\mathbb{E}[X \mid Z])\right\|^2\right], \quad \text{(Substitute in definitions of } X', Y'.)$$

$$= \mathbb{E}\left[\left\|a(Y - \mathbf{h}^\top X) + b(\mathbb{E}[Y \mid Z] - \mathbf{h}^\top\mathbb{E}[X \mid Z])\right\|^2\right], \quad\quad \text{(Distribute the subtraction.)}$$

$$= a^2\mathbb{E}\left[\left\|Y - \mathbf{h}^\top X\right\|^2\right] + b^2\mathbb{E}\left[\left\|\mathbb{E}[Y \mid Z] - \mathbf{h}^\top\mathbb{E}[X \mid Z]\right\|^2\right] \quad\quad \text{(Expand squared norm.)}$$

$$+ 2ab\mathbb{E}\left[\left(Y - \mathbf{h}^\top X\right)^\top\left(\mathbb{E}[Y \mid Z] - \mathbf{h}^\top\mathbb{E}[X \mid Z]\right)\right]. \tag{17}$$

First we note that from the definitions of $a, b$ we have

$$a^2 = \sqrt{\alpha}, \qquad b^2 + 2ab = \left(\sqrt{1+\alpha} - \sqrt{\alpha}\right)^2 + 2\sqrt{\alpha}\left(\sqrt{1+\alpha} - \sqrt{\alpha}\right) = 1. \tag{18}$$

Now we evaluate the cross term in Eq. (17)

$$\Rightarrow \mathbb{E}\left[\left(Y - \mathbf{h}^\top X\right)^\top\left(\mathbb{E}[Y \mid Z] - \mathbf{h}^\top\mathbb{E}[X \mid Z]\right)\right]$$

$$= \mathbb{E}\left[\mathbb{E}\left[\left(Y - \mathbf{h}^\top X\right)^\top\left(\mathbb{E}[Y \mid Z] - \mathbf{h}^\top\mathbb{E}[X \mid Z]\right) \mid Z\right]\right], \quad\quad \text{(Law of iterated expectation.)}$$

$$= \mathbb{E}\left[\mathbb{E}\left[\left(Y - \mathbf{h}^\top X\right)^\top \mid Z\right]\left(\mathbb{E}[Y \mid Z] - \mathbf{h}^\top\mathbb{E}[X \mid Z]\right)\right] \quad \text{(Taking out what is known; Eq. (15).)}$$

$$= \mathbb{E}\left[\left(\mathbb{E}[Y \mid Z] - \mathbf{h}^\top\mathbb{E}[X \mid Z]\right)^\top\left(\mathbb{E}[Y \mid Z] - \mathbf{h}^\top\mathbb{E}[X \mid Z]\right)\right]$$

$$= \mathbb{E}\left[\left\|\mathbb{E}[Y \mid Z] - \mathbf{h}^\top\mathbb{E}[X \mid Z]\right\|^2\right].$$

Substituting this back in Eq. (17) we get

$$\Rightarrow \mathbb{E}\left[\left\|Y' - \mathbf{h}^\top X'\right\|^2\right]$$

$$= a^2\mathbb{E}\left[\left\|Y - \mathbf{h}^\top X\right\|^2\right] + (b^2 + 2ab)\mathbb{E}\left[\left\|\mathbb{E}[Y \mid Z] - \mathbf{h}^\top\mathbb{E}[X \mid Z]\right\|^2\right],$$

$$= \alpha\mathbb{E}\left[\left\|Y - \mathbf{h}^\top X\right\|^2\right] + \mathbb{E}\left[\left\|\mathbb{E}[Y \mid Z] - \mathbf{h}^\top\mathbb{E}[X \mid Z]\right\|^2\right], \quad\quad \text{(From Eq. (18).)}$$

$$= \alpha R_{\text{ERM}}^{\mathfrak{M}}(\mathbf{h}) + R_{\text{IV}}^{\mathfrak{M}}(\mathbf{h}) - \mathbb{E}[\text{Var}(Y \mid Z)], \quad\quad \text{(From Eq. (13).)}$$

$$= R_{\text{IVL}_\alpha}^{\mathfrak{M}}(\mathbf{h}) - \mathbb{E}[\text{Var}(Y \mid Z)].$$

$\square$

## F.2 Proof of Proposition 2—Existence of an interventional distribution given a DA

**Proposition 2** (unique stationary interventional distribution). *In SEM $\mathfrak{A}$ from Eq. (9), given any $(\mathbf{g}, \mathbf{c}, \mathbf{n}_X, \mathbf{n}_Y) \sim P_{G,C,N_X,N_Y}^{\mathfrak{A}}$, if for all $(\mathbf{x}_0, \mathbf{y}_0) \in \mathcal{X} \times \mathcal{Y}$ the unique limits*

$$\mathbf{x}^{\mathfrak{A}} := \lim_{t \to \infty} \mathbf{x}_t^{\mathfrak{A}} = \lim_{t \to \infty} \tau\big(\mathbf{y}_{t-1}^{\mathfrak{A}}, \mathbf{c}, \mathbf{n}_X\big),$$

$$\mathbf{y}^{\mathfrak{A}} := \lim_{t \to \infty} \mathbf{y}_t^{\mathfrak{A}} = \lim_{t \to \infty} f\big(\mathbf{x}_{t-1}^{\mathfrak{A}}\big) + \epsilon(\mathbf{c}) + \mathbf{n}_Y$$

*exist, then in $\mathfrak{A}; \mathrm{do}(\tau := \mathbf{g}\tau)$ the unique limits*

$$\mathbf{x}^{\mathfrak{A};\mathrm{do}(\tau:=\mathbf{g}\tau)} := \lim_{t \to \infty} \mathbf{x}_t^{\mathfrak{A};\mathrm{do}(\tau:=\mathbf{g}\tau)} = \lim_{t \to \infty} \mathbf{g}\tau\Big(\mathbf{y}_{t-1}^{\mathfrak{A};\mathrm{do}(\tau:=\mathbf{g}\tau)}, \mathbf{c}, \mathbf{n}_X\Big) = \mathbf{g}\mathbf{x}^{\mathfrak{A}},$$

$$\mathbf{y}^{\mathfrak{A};\mathrm{do}(\tau:=\mathbf{g}\tau)} := \lim_{t \to \infty} \mathbf{y}_t^{\mathfrak{A};\mathrm{do}(\tau:=\mathbf{g}\tau)} = \lim_{t \to \infty} f\Big(\mathbf{x}_{t-1}^{\mathfrak{A};\mathrm{do}(\tau:=\mathbf{g}\tau)}\Big) + \epsilon(\mathbf{c}) + \mathbf{n}_Y = \mathbf{y}^{\mathfrak{A}}$$

*also exist.*

*Proof.* First we try to show that

$$\mathbf{y}_t^{\mathfrak{A};\mathrm{do}(\tau:=\mathbf{g}\tau)} = \mathbf{y}_t^{\mathfrak{A}}. \tag{19}$$

For the base case, we have by construction

$$\mathbf{y}_0^{\mathfrak{A};\mathrm{do}(\tau:=\mathbf{g}\tau)} := \mathbf{y}_0 =: \mathbf{y}_0^{\mathfrak{A}}.$$

For the step case, assuming that $\mathbf{y}_t^{\mathfrak{A};\mathrm{do}(\tau:=\mathbf{g}\tau)} = \mathbf{y}_t^{\mathfrak{A}}$, we have[13],

$$
\begin{aligned}
\mathbf{y}_{t+2}^{\mathfrak{A};\mathrm{do}(\tau:=\mathbf{g}\tau)} &= f\Big(\mathbf{x}_{t+1}^{\mathfrak{A};\mathrm{do}(\tau:=\mathbf{g}\tau)}\Big) + \epsilon(\mathbf{c}) + \mathbf{n}_Y, \\
&= f\Big(\mathbf{g}\tau\Big(\mathbf{y}_t^{\mathfrak{A};\mathrm{do}(\tau:=\mathbf{g}\tau)}, \mathbf{c}, \mathbf{n}_X\Big)\Big) + \epsilon(\mathbf{c}) + \mathbf{n}_Y, \\
&= f\big(\tau\big(\mathbf{y}_t^{\mathfrak{A};\mathrm{do}(\tau:=\mathbf{g}\tau)}, \mathbf{c}, \mathbf{n}_X\big)\big) + \epsilon(\mathbf{c}) + \mathbf{n}_Y, && \text{(Invariance of } f \text{ to } \mathbf{g}.) \\
&= f\big(\tau\big(\mathbf{y}_t^{\mathfrak{A}}, \mathbf{c}, \mathbf{n}_X\big)\big) + \epsilon(\mathbf{c}) + \mathbf{n}_Y, && \text{(Assumption } \mathbf{y}_t^{\mathfrak{A};\mathrm{do}(\tau:=\mathbf{g}\tau)} = \mathbf{y}_t^{\mathfrak{A}}.) \\
&= f\big(\mathbf{x}_{t+1}^{\mathfrak{A}}\big) + \epsilon(\mathbf{c}) + \mathbf{n}_Y, \\
&= \mathbf{y}_{t+2}^{\mathfrak{A}}.
\end{aligned}
$$

Hence, we have shown that Eq. (19) holds for all even $t$. For odd $t$, we simply replace $t = 0$ with $t = 1$ in the base case

$$
\begin{aligned}
\mathbf{y}_1^{\mathfrak{A};\mathrm{do}(\tau:=\mathbf{g}\tau)} &= f\Big(\mathbf{x}_0^{\mathfrak{A};\mathrm{do}(\tau:=\mathbf{g}\tau)}\Big) + \epsilon(\mathbf{c}) + \mathbf{n}_Y, \\
&= f\big(\mathbf{x}_0^{\mathfrak{A}}\big) + \epsilon(\mathbf{c}) + \mathbf{n}_Y, && \text{(Definitions } \mathbf{x}_0^{\mathfrak{A};\mathrm{do}(\tau:=\mathbf{g}\tau)} := \mathbf{x}_0 =: \mathbf{x}_0^{\mathfrak{A}}.) \\
&= \mathbf{y}_1^{\mathfrak{A}},
\end{aligned}
$$

We have now finally shown that Eq. (19) holds for all $t \geq 0$.

Next, it is now relatively straightforward to show that for any $t > 0$, we have

$$
\begin{aligned}
\mathbf{x}_t^{\mathfrak{A};\mathrm{do}(\tau:=\mathbf{g}\tau)} &= \mathbf{g}\tau\Big(\mathbf{y}_{t-1}^{\mathfrak{A};\mathrm{do}(\tau:=\mathbf{g}\tau)}, \mathbf{c}, \mathbf{n}_X\Big), \\
&= \mathbf{g}\tau\big(\mathbf{y}_{t-1}^{\mathfrak{A}}, \mathbf{c}, \mathbf{n}_X\big), && \text{(Follows from Eq. (19).)} \\
&= \mathbf{g}\mathbf{x}_t^{\mathfrak{A}}. \tag{20}
\end{aligned}
$$

Finally, by applying limit as $t \to \infty$ to both sides of Eq. (19) and Eq. (20), we get

$$\mathbf{y}^{\mathfrak{A};\mathrm{do}(\tau:=\mathbf{g}\tau)} = \lim_{t \to \infty} \mathbf{y}_t^{\mathfrak{A};\mathrm{do}(\tau:=\mathbf{g}\tau)} = \lim_{t \to \infty} \mathbf{y}_t^{\mathfrak{A}} = \mathbf{y}^{\mathfrak{A}},$$

$$\mathbf{x}^{\mathfrak{A};\mathrm{do}(\tau:=\mathbf{g}\tau)} = \lim_{t \to \infty} \mathbf{x}_t^{\mathfrak{A};\mathrm{do}(\tau:=\mathbf{g}\tau)} = \lim_{t \to \infty} \mathbf{g}\mathbf{x}_t^{\mathfrak{A}} = \mathbf{g} \lim_{t \to \infty} \mathbf{x}_t^{\mathfrak{A}} = \mathbf{g}\mathbf{x}^{\mathfrak{A}}, \tag{21}$$

where the limit can be moved past $\mathbf{g}$ in Eq. (21) because $\mathbf{g}$ is assumed continuous in its domain.

$\square$

---

[13]Note that here the step size for proof by induction would be $\Delta t = 2$ since $\mathbf{y}_t$ precedes $\mathbf{y}_{t+2}$. Similar is the case for $\mathbf{x}_t$ as well.

### F.3 Proof of Theorem 1—Robust prediction with IVL regression

**Theorem 1** (robust prediction with IVL regression). *For SEM $\mathfrak{M}$ in Example 1, the following holds:*

$$\hat{\mathbf{h}}_{IVL_\alpha}^{\mathfrak{M}} \in \underset{\mathbf{h}}{\arg\min} \max_{\zeta \in \mathcal{P}_\alpha} R_{ERM}^{\mathfrak{M};\mathrm{do}(\mathbf{\Gamma}^\top(\cdot):=\zeta)}(\mathbf{h}), \quad s.t. \quad \mathcal{P}_\alpha := \left\{ \zeta \,\middle|\, \zeta\zeta^\top \preccurlyeq \left(\frac{1}{\alpha}+1\right)\mathbf{\Gamma}^\top \mathbf{\Sigma}_Z^{\mathfrak{M}} \mathbf{\Gamma} \right\}.$$

*Proof.* Write $X$ in terms of the exogenous variables $C, Z, N_X, N_Y$ using the reduced form from Lemma 3 as

$$X = \tilde{Z} + \tilde{C} + \tilde{N}, \tag{22}$$

where for readability we represent

$$\tilde{Z} := \mathbf{M}_{m \times m} \mathbf{\Gamma}^\top Z, \qquad \tilde{C} := \mathbf{M}\begin{bmatrix}\mathbf{T}^\top \\ \boldsymbol{\epsilon}^\top\end{bmatrix} C, \qquad \tilde{N} := \sigma \cdot \mathbf{M}\begin{bmatrix}N_X \\ N_Y\end{bmatrix},$$

with

$$\mathbf{M} := \begin{bmatrix}\mathbf{M}_{m \times m} & \mathbf{M}_{m \times 1} \\ \mathbf{M}_{1 \times m} & \mathbf{M}_{1 \times 1}\end{bmatrix} = \begin{bmatrix}\mathbf{I}_m & -\boldsymbol{\tau}^\top \\ -\mathbf{f}^\top & 1\end{bmatrix}^{-1}.$$

Now, we start by writing the ERM objective under the intervention $\mathrm{do}(\mathbf{\Gamma}^\top(\cdot):=\zeta)$ as

$$\Rightarrow R_{ERM}^{\mathfrak{M};\mathrm{do}(\mathbf{\Gamma}^\top(\cdot):=\zeta)}(\mathbf{h})$$

$$= \mathbb{E}^{\mathfrak{M};\mathrm{do}(\mathbf{\Gamma}^\top(\cdot):=\zeta)}\left[\left\|Y - \mathbf{h}^\top X\right\|^2\right],$$

$$= \mathbb{E}^{\mathfrak{M};\mathrm{do}(\mathbf{\Gamma}^\top(\cdot):=\zeta)}\left[\left\|\xi + (\mathbf{f}-\mathbf{h})^\top\left(\tilde{Z}+\tilde{C}+\tilde{N}\right)\right\|^2\right], \qquad (Y \text{ structural form \& Eq. (22).})$$

$$= \mathbb{E}^{\mathfrak{M};\mathrm{do}(\mathbf{\Gamma}^\top(\cdot):=\zeta)}\left[\left\|\xi + (\mathbf{f}-\mathbf{h})^\top\left(\mathbf{M}_{m \times m}\zeta+\tilde{C}+\tilde{N}\right)\right\|^2\right], \qquad (\tilde{Z} \text{ \& intervention definition.})$$

$$= \mathbb{E}^{\mathfrak{M};\mathrm{do}(\mathbf{\Gamma}^\top(\cdot):=\zeta)}\left[\left\|\xi + (\mathbf{f}-\mathbf{h})^\top\left(\tilde{C}+\tilde{N}\right)+(\mathbf{f}-\mathbf{h})^\top\mathbf{M}_{m \times m}\zeta\right\|^2\right],$$

$$= \mathbb{E}^{\mathfrak{M};\mathrm{do}(\mathbf{\Gamma}^\top(\cdot):=\zeta)}\left[\left\|\xi + (\mathbf{f}-\mathbf{h})^\top\left(\tilde{C}+\tilde{N}\right)+\mathbf{h'}^\top\zeta\right\|^2\right], \quad (\text{Define } \mathbf{h'}^\top := (\mathbf{f}-\mathbf{h})^\top\mathbf{M}_{m \times m}.)$$

$$= \mathbb{E}^{\mathfrak{M};\mathrm{do}(\mathbf{\Gamma}^\top(\cdot):=\zeta)}\left[\left\|\xi + (\mathbf{f}-\mathbf{h})^\top\left(\tilde{C}+\tilde{N}\right)\right\|^2\right] + \mathbb{E}^{\mathfrak{M};\mathrm{do}(\mathbf{\Gamma}^\top(\cdot):=\zeta)}\left[\left\|\mathbf{h'}^\top\zeta\right\|^2\right],$$

$$\qquad (\text{Follows from exogeneity of } \zeta \text{ under intervention, } \Rightarrow \text{cross term zeros-out.})$$

$$= \mathbb{E}^{\mathfrak{M};\mathrm{do}(\mathbf{\Gamma}^\top(\cdot):=\mathbf{0}_m)}\left[\left\|Y - \mathbf{h}^\top X\right\|^2\right] + \mathbb{E}^{\mathfrak{M};\mathrm{do}(\mathbf{\Gamma}^\top(\cdot):=\zeta)}\left[\left\|\mathbf{h'}^\top\zeta\right\|^2\right], \tag{23}$$

$$= \mathbb{E}^{\mathfrak{M};\mathrm{do}(\mathbf{\Gamma}^\top(\cdot):=\mathbf{0}_m)}\left[\left\|Y - \mathbf{h}^\top X\right\|^2\right] + \left\|\mathbf{h'}^\top\zeta\right\|^2,$$

$$= \mathbb{E}^{\mathfrak{M};\mathrm{do}(\mathbf{\Gamma}^\top(\cdot):=\mathbf{0}_m)}\left[\left\|Y - \mathbf{h}^\top X\right\|^2\right] + \mathrm{tr}\left(\zeta^\top\mathbf{h'}\mathbf{h'}^\top\zeta\right),$$

$$= \mathbb{E}^{\mathfrak{M};\mathrm{do}(\mathbf{\Gamma}^\top(\cdot):=\mathbf{0}_m)}\left[\left\|Y - \mathbf{h}^\top X\right\|^2\right] + \mathrm{tr}\left(\mathbf{h'}^\top\zeta\zeta^\top\mathbf{h'}\right). \tag{24}$$

Now, note that the maximum of the trace term over $\zeta \in \mathcal{P}_\alpha$ gives

$$\Rightarrow \max_{\zeta \in \mathcal{P}_\alpha} \mathrm{tr}\left(\mathbf{h'}^\top\zeta\zeta^\top\mathbf{h'}\right),$$

$$= \left(\frac{1}{\alpha}+1\right)\mathrm{tr}\left(\mathbf{h'}^\top\left(\mathbf{\Gamma}^\top\mathbb{E}^{\mathfrak{M}}\left[ZZ^\top\right]\mathbf{\Gamma}\right)\mathbf{h'}\right), \qquad (\text{Linearity of trace and definition of } \mathcal{P}_\alpha.)$$

$$= \left(\frac{1}{\alpha}+1\right)\mathbb{E}^{\mathfrak{M}}\left[\mathrm{tr}\left(\mathbf{h'}^\top\mathbf{\Gamma}^\top ZZ^\top\mathbf{\Gamma}\mathbf{h'}\right)\right], \qquad (\text{Linearity of expectation.})$$

$$= \left(\frac{1}{\alpha}+1\right)\mathbb{E}^{\mathfrak{M}}\left[\mathrm{tr}\left(Z^\top\mathbf{\Gamma}\mathbf{h'}\mathbf{h'}^\top\mathbf{\Gamma}^\top Z\right)\right], \qquad (\text{Cyclic property of trace.})$$

$$= \left(\frac{1}{\alpha} + 1\right) \mathbb{E}^{\mathfrak{M}}\left[\left\|\mathbf{h}'^{\top}\mathbf{\Gamma}^{\top}Z\right\|^2\right],$$

$$= \left(\frac{1}{\alpha} + 1\right) \mathbb{E}^{\mathfrak{M}}\left[\left\|(\mathbf{f} - \mathbf{h})^{\top}\mathbf{M}_{m \times m}\mathbf{\Gamma}^{\top}Z\right\|^2\right], \qquad \text{(Substitute in definition of } \mathbf{h}'^{\top}.)$$

$$= \left(\frac{1}{\alpha} + 1\right) \mathbb{E}^{\mathfrak{M}}\left[\left\|(\mathbf{f} - \mathbf{h})^{\top}\tilde{Z}\right\|^2\right]. \qquad \text{(Definition of } \tilde{Z}.)$$

We can now substitute this in while maximizing both sides of Eq. (24) over interventions $\zeta \in \mathcal{P}_\alpha$ as

$$\Rightarrow \max_{\zeta \in \mathcal{P}_\alpha} R_{\text{ERM}}^{\mathfrak{M};\text{do}\left(\mathbf{\Gamma}^{\top}(\cdot):=\mathbf{0}_m\right)}(\mathbf{h})$$

$$= \mathbb{E}^{\mathfrak{M};\text{do}\left(\mathbf{\Gamma}^{\top}(\cdot):=\mathbf{0}_m\right)}\left[\left\|Y - \mathbf{h}^{\top}X\right\|^2\right] + \max_{\zeta \in \mathcal{P}_\alpha} \text{tr}\left(\mathbf{h}'^{\top}\zeta\zeta^{\top}\mathbf{h}'\right), \qquad \text{(First term does not have } \zeta.)$$

$$= \mathbb{E}^{\mathfrak{M};\text{do}\left(\mathbf{\Gamma}^{\top}(\cdot):=\mathbf{0}_m\right)}\left[\left\|Y - \mathbf{h}^{\top}X\right\|^2\right] + \left(\frac{1}{\alpha} + 1\right) \mathbb{E}^{\mathfrak{M}}\left[\left\|(\mathbf{f} - \mathbf{h})^{\top}\tilde{Z}\right\|^2\right],$$

$$= \mathbb{E}^{\mathfrak{M}}\left[\left\|Y - \mathbf{h}^{\top}X\right\|^2\right] + \frac{1}{\alpha} \mathbb{E}^{\mathfrak{M}}\left[\left\|(\mathbf{f} - \mathbf{h})^{\top}\tilde{Z}\right\|^2\right], \qquad \text{(Inverse step of Eq. (23).)}$$

$$= \mathbb{E}^{\mathfrak{M}}\left[\left\|Y - \mathbf{h}^{\top}X\right\|^2\right] + \frac{1}{\alpha} \mathbb{E}^{\mathfrak{M}}\left[\left\|(\mathbf{f} - \mathbf{h})^{\top}\mathbb{E}[X \mid Z]\right\|^2\right], \qquad \text{(From conditional exp. of Eq. (22).)}$$

$$= \mathbb{E}^{\mathfrak{M}}\left[\left\|Y - \mathbf{h}^{\top}X\right\|^2\right] + \frac{1}{\alpha} \mathbb{E}^{\mathfrak{M}}\left[\left\|\mathbb{E}\left[\mathbf{f}^{\top}X \mid Z\right] - \mathbf{h}^{\top}\mathbb{E}[X \mid Z]\right\|^2\right], \qquad \text{(Linearity of expectation.)}$$

$$= \mathbb{E}^{\mathfrak{M}}\left[\left\|Y - \mathbf{h}^{\top}X\right\|^2\right] + \frac{1}{\alpha} \mathbb{E}^{\mathfrak{M}}\left[\left\|\mathbb{E}[Y \mid Z] - \mathbf{h}^{\top}\mathbb{E}[X \mid Z]\right\|^2\right], \qquad \text{(Inverse step of Eq. (23).)}$$

$$= R_{\text{ERM}}^{\mathfrak{M}}(\mathbf{h}) + \frac{1}{\alpha}\left(R_{\text{IV}}^{\mathfrak{M}}(\mathbf{h}) - \mathbb{E}[\text{Var}(Y \mid Z)]\right), \qquad \text{(From Eq. (13).)}$$

$$= \frac{1}{\alpha}\left(R_{\text{IVL}_\alpha}^{\mathfrak{M}}(\mathbf{h}) - \mathbb{E}[\text{Var}(Y \mid Z)]\right).$$

$$\square$$

## F.4 Proof of Theorem 2—Causal estimation with IVL regression

**Theorem 2** (causal estimation with IVL regression). *In SEM $\mathfrak{M}$ of Example 1, for $\alpha < \infty$, we have*

$$\mathrm{CER}_{\mathfrak{M}}\left(\hat{\mathbf{h}}_{IVL_{\alpha}}^{\mathfrak{M}}\right) \leq \mathrm{CER}_{\mathfrak{M}}\left(\hat{\mathbf{h}}_{ERM}^{\mathfrak{M}}\right), \qquad \textit{equality iff} \qquad \mathbb{E}^{\mathfrak{M}}[X \mid Z] \perp_{\text{a.s.}} \mathbb{E}^{\mathfrak{M}}[X \mid \xi].$$

*Proof.* For $\hat{\mathbf{h}}_{\mathrm{IVL}_{\alpha}}^{\mathfrak{M}}$, we have from Proposition 1

$$\left\|\hat{\mathbf{h}}_{\mathrm{IVL}_{\alpha}}^{\mathfrak{M}} - \mathbf{f}\right\|_{\boldsymbol{\Sigma}_X^{\mathfrak{M}}}^2 = \left\|\mathbb{E}\left[X'X'^{\top}\right]^{-1}\mathbb{E}\left[X'Y'^{\top}\right] - \mathbf{f}\right\|_{\boldsymbol{\Sigma}_X^{\mathfrak{M}}}^2.$$

Note that we have

$$
\begin{aligned}
&\Rightarrow \mathbb{E}\left[X'Y'^{\top}\right] \\
&= \mathbb{E}\left[X'(aY + b\mathbb{E}[Y \mid Z])^{\top}\right], \\
&= \mathbb{E}\left[X'(aY + b\mathbb{E}\left[\mathbf{f}^{\top}X + \xi \mid Z\right])^{\top}\right], \\
&= \mathbb{E}\left[X'(aY + b\mathbf{f}^{\top}\mathbb{E}[X \mid Z])^{\top}\right], && \text{(Dy definition } Z \perp\!\!\!\perp \xi.\text{)} \\
&= \mathbb{E}\left[X'(a\mathbf{f}^{\top}X + a\xi + b\mathbf{f}^{\top}\mathbb{E}[X \mid Z])^{\top}\right], \\
&= \mathbb{E}\left[X'(\mathbf{f}^{\top}X' + a\xi)^{\top}\right], && \text{(Substituting in } X' := aX + b\mathbb{E}[X \mid Z].\text{)} \\
&= \mathbb{E}\left[X'X'^{\top}\mathbf{f} + aX'\xi^{\top}\right], \\
&= \mathbb{E}\left[X'X'^{\top}\right]\mathbf{f} + a\mathbb{E}\left[X'\xi^{\top}\right], \\
&= \mathbb{E}\left[X'X'^{\top}\right]\mathbf{f} + a^2\mathbb{E}\left[X\xi^{\top}\right], && (Z \perp\!\!\!\perp \xi, \text{ therefore } \mathbb{E}\left[X'\xi^{\top}\right] = a\mathbb{E}\left[X\xi^{\top}\right].) \\
&= \mathbb{E}\left[X'X'^{\top}\right]\mathbf{f} + \alpha\mathbb{E}\left[X\xi^{\top}\right], && (25)
\end{aligned}
$$

We also see that

$$
\begin{aligned}
&\Rightarrow \mathbb{E}\left[X'X'^{\top}\right] \\
&= \mathbb{E}\left[(aX + b\mathbb{E}[X \mid Z])(aX + b\mathbb{E}[X \mid Z])^{\top}\right], \\
&= \mathbb{E}\left[\left(aX + b\tilde{Z}\right)\left(aX + b\tilde{Z}\right)^{\top}\right], && \text{(Set } \tilde{Z} := \mathbb{E}[X \mid Z] \text{ for brevity.)} \\
&= a^2\mathbb{E}\left[XX^{\top}\right] + b^2\mathbb{E}\left[\tilde{Z}\tilde{Z}^{\top}\right] + ab\mathbb{E}\left[X\tilde{Z}^{\top}\right] + ab\mathbb{E}\left[\tilde{Z}X^{\top}\right], \\
&= a^2\mathbb{E}\left[XX^{\top}\right] + \left(b^2 + 2ab\right)\boldsymbol{\Sigma}_{\tilde{Z}}, && \text{(Because } \mathbb{E}\left[X\tilde{Z}^{\top}\right] = \boldsymbol{\Sigma}_{\tilde{Z}}.\text{)} \\
&= \alpha\mathbb{E}\left[XX^{\top}\right] + \boldsymbol{\Sigma}_{\tilde{Z}}, && (26)
\end{aligned}
$$

where we substituted in Eq. (18) in Eq. (26).

Finally, we now have

$$
\begin{aligned}
&\Rightarrow \left\|\hat{\mathbf{h}}_{\mathrm{IVL}_{\alpha}}^{\mathfrak{M}} - \mathbf{f}\right\|_{\boldsymbol{\Sigma}_X^{\mathfrak{M}}}^2 \\
&= \left\|\mathbb{E}\left[X'X'^{\top}\right]^{-1}\mathbb{E}\left[X'Y'^{\top}\right] - \mathbf{f}\right\|_{\boldsymbol{\Sigma}_X^{\mathfrak{M}}}^2, \\
&= \left\|\mathbb{E}\left[X'X'^{\top}\right]^{-1}\left(\mathbb{E}\left[X'X'^{\top}\right]\mathbf{f} + \alpha\mathbb{E}\left[X\xi^{\top}\right]\right) - \mathbf{f}\right\|_{\boldsymbol{\Sigma}_X^{\mathfrak{M}}}^2, && \text{(Substituting in Eq. (25).)} \\
&= \left\|\mathbf{f} + \alpha\mathbb{E}\left[X'X'^{\top}\right]^{-1}\mathbb{E}\left[X\xi^{\top}\right] - \mathbf{f}\right\|_{\boldsymbol{\Sigma}_X^{\mathfrak{M}}}^2,
\end{aligned}
$$

$$= \left\| \alpha \mathbb{E}\left[X'X'^\top\right]^{-1} \mathbb{E}\left[X\xi^\top\right] \right\|^2_{\mathbf{\Sigma}^{\mathfrak{M}}_X},$$

$$= \left\| \alpha \left(\alpha \mathbb{E}\left[XX^\top\right] + \mathbf{\Sigma}_{\tilde{Z}}\right)^{-1} \mathbb{E}\left[X\xi^\top\right] \right\|^2_{\mathbf{\Sigma}^{\mathfrak{M}}_X}, \qquad\qquad \text{(Substituting in Eq. (26).)}$$

$$= \left\| \left(\mathbf{S}^\top\mathbf{S} + \frac{1}{\alpha}\mathbf{S}^\top\mathbf{D}\mathbf{S}\right)^{-1} \mathbb{E}\left[X\xi^\top\right] \right\|^2_{\mathbf{S}^\top\mathbf{S}}, \qquad\qquad \text{(Using Lemma 2.)}$$

$$= \left\| \mathbf{S}^{-1}\left(\mathbf{I}_m + \frac{1}{\alpha}\mathbf{D}\right)^{-1} \mathbf{S}^{-\top} \mathbb{E}\left[X\xi^\top\right] \right\|^2_{\mathbf{S}^\top\mathbf{S}}, \qquad\qquad \text{($\mathbf{S}$ is invertible.)}$$

$$= \left\| \left(\mathbf{I}_m + \frac{1}{\alpha}\mathbf{D}\right)^{-1} \mathbf{S}^{-\top} \mathbb{E}\left[X\xi^\top\right] \right\|^2, \qquad\qquad \text{(Switch to $\ell_2$ norm.)}$$

$$\leq \left\| \mathbf{S}^{-\top} \mathbb{E}\left[X\xi^\top\right] \right\|^2, \qquad\qquad\qquad\qquad\qquad\qquad\qquad (27)$$

$$= \left\| \mathbf{S}\mathbf{S}^{-1}\mathbf{S}^{-\top} \mathbb{E}\left[X\xi^\top\right] \right\|^2, \qquad\qquad \text{(Substituting $\mathbf{I} = \mathbf{S}\mathbf{S}^{-1}$.)}$$

$$= \left\| \mathbf{S}^{-1}\mathbf{S}^{-\top} \mathbb{E}\left[X\xi^\top\right] \right\|^2_{\mathbf{S}^\top\mathbf{S}}, \qquad\qquad \text{(Back to weighted norm.)}$$

$$= \left\| \mathbb{E}\left[XX^\top\right]^{-1} \mathbb{E}\left[X\xi^\top\right] \right\|^2_{\mathbf{\Sigma}^{\mathfrak{M}}_X}, \qquad\qquad \text{(Substituting $\mathbf{\Sigma}^{\mathfrak{M}}_X := \mathbb{E}^{\mathfrak{M}}\left[XX^\top\right] = \mathbf{S}^\top\mathbf{S}$.)}$$

$$= \left\| \mathbf{f} + \mathbb{E}\left[XX^\top\right]^{-1} \mathbb{E}\left[X\xi^\top\right] - \mathbf{f} \right\|^2_{\mathbf{\Sigma}^{\mathfrak{M}}_X}, \qquad\qquad \text{(Adding and subtracting $\mathbf{f}$.)}$$

$$= \left\| \mathbb{E}\left[XX^\top\right]^{-1} \left(\mathbb{E}\left[XX^\top\right]\mathbf{f} + \mathbb{E}\left[X\xi^\top\right]\right) - \mathbf{f} \right\|^2_{\mathbf{\Sigma}^{\mathfrak{M}}_X}, \quad \text{(Substitute $\mathbf{I} = \mathbb{E}\left[XX^\top\right]^{-1}\mathbb{E}\left[XX^\top\right]$.)}$$

$$= \left\| \mathbb{E}\left[XX^\top\right]^{-1} \mathbb{E}\left[X\left(\mathbf{f}^\top X + \xi\right)^\top\right] - \mathbf{f} \right\|^2_{\mathbf{\Sigma}^{\mathfrak{M}}_X}, \qquad\qquad \text{(Linearity of expectation.)}$$

$$= \left\| \mathbb{E}\left[XX^\top\right]^{-1} \mathbb{E}\left[XY^\top\right] - \mathbf{f} \right\|^2_{\mathbf{\Sigma}^{\mathfrak{M}}_X}, \qquad\qquad \text{(Substituting $Y = \mathbf{f}^\top X + \xi$.)}$$

$$= \left\| \hat{\mathbf{h}}^{\mathfrak{M}}_{\text{ERM}} - \mathbf{f} \right\|^2_{\mathbf{\Sigma}^{\mathfrak{M}}_X}, \qquad\qquad \text{(Closed form ERM solution.)}$$

where inequality Eq. (27) holds because $\mathbf{D}$ is non-negative diagonal. Furthermore, inequality Eq. (27) only holds with equality iff $\mathbf{S}^{-\top}\mathbb{E}\left[X\xi^\top\right]$ is in the kernel of $\mathbf{D}$. Or equivalently, iff $\mathbb{E}\left[X\xi^\top\right]$ is in the kernel of $\mathbf{S}^\top\mathbf{D}\mathbf{S} = \mathbf{\Sigma}_{\tilde{Z}}$, which from Lemma 1 is true iff

$$\mathbb{E}^{\mathfrak{M}}[X \mid Z] \quad \perp \quad \mathbb{E}^{\mathfrak{M}}[X \mid \xi] \qquad \text{a.s.}$$

$\square$

## F.5 Proof of Theorem 3—Causal estimation with DA+ERM

**Theorem 3** (causal estimation with DA+ERM). *For SEM $\mathfrak{A}$ in Example 2, the following holds:*

$$\mathrm{CER}_{\mathfrak{A}}\left(\hat{\mathbf{h}}_{DA_G+ERM}^{\mathfrak{A}}\right) \leq \mathrm{CER}_{\mathfrak{A}}\left(\hat{\mathbf{h}}_{ERM}^{\mathfrak{A}}\right), \qquad \text{equality iff} \qquad \mathbb{E}^{\mathfrak{A}}[GX \mid G] \perp_{\mathrm{a.s.}} \mathbb{E}^{\mathfrak{A}}[X \mid \xi].$$

*Proof.* We have

$$\Rightarrow \left\|\hat{\mathbf{h}}_{\mathrm{DA}_G+\mathrm{ERM}}^{\mathfrak{A}} - \mathbf{f}\right\|_{\boldsymbol{\Sigma}_X^{\mathfrak{A}}}$$

$$= \left\|\mathbb{E}\left[(GX)(GX)^{\top}\right]^{-1}\mathbb{E}\left[(GX)Y^{\top}\right] - \mathbf{f}\right\|_{\boldsymbol{\Sigma}_X^{\mathfrak{A}}},$$

$$= \left\|\mathbb{E}\left[(GX)(GX)^{\top}\right]^{-1}\mathbb{E}\left[(GX)\left(\mathbf{f}^{\top}X + \xi\right)^{\top}\right] - \mathbf{f}\right\|_{\boldsymbol{\Sigma}_X^{\mathfrak{A}}}, \qquad \text{(Structural eq. of } Y.)$$

$$= \left\|\mathbb{E}\left[(GX)(GX)^{\top}\right]^{-1}\mathbb{E}\left[(GX)\left(\mathbf{f}^{\top}(GX) + \xi\right)^{\top}\right] - \mathbf{f}\right\|_{\boldsymbol{\Sigma}_X^{\mathfrak{A}}}, \qquad \text{(Using } \mathcal{G}\text{-invariance of } \mathbf{f}.)$$

$$= \left\|\left(\mathbf{f} + \mathbb{E}\left[(GX)(GX)^{\top}\right]^{-1}\mathbb{E}\left[(GX)\xi^{\top}\right]\right) - \mathbf{f}\right\|_{\boldsymbol{\Sigma}_X^{\mathfrak{A}}},$$

$$= \left\|\mathbb{E}\left[(GX)(GX)^{\top}\right]^{-1}\mathbb{E}\left[(GX)\xi^{\top}\right]\right\|_{\boldsymbol{\Sigma}_X^{\mathfrak{A}}},$$

$$= \left\|\mathbb{E}\left[\left(X + \tilde{G}\right)\left(X + \tilde{G}\right)^{\top}\right]^{-1}\mathbb{E}\left[\left(X + \tilde{G}\right)\xi^{\top}\right]\right\|_{\boldsymbol{\Sigma}_X^{\mathfrak{A}}}, \qquad \text{(Let } \tilde{G} := \mathbb{E}[GX \mid G] = \gamma \cdot \boldsymbol{\Gamma}^{\top}G.)$$

$$= \left\|\left(\mathbb{E}\left[XX^{\top}\right] + \mathbb{E}\left[\tilde{G}\tilde{G}^{\top}\right]\right)^{-1}\mathbb{E}\left[X\xi^{\top}\right]\right\|_{\boldsymbol{\Sigma}_X^{\mathfrak{A}}}, \qquad \text{(Using } \tilde{G} \perp\!\!\!\perp X, \xi.)$$

$$= \left\|\left(\mathbf{S}^{\top}\mathbf{S} + \mathbf{S}^{\top}\mathbf{D}\mathbf{S}\right)^{-1}\mathbb{E}\left[X\xi^{\top}\right]\right\|_{\mathbf{S}^{\top}\mathbf{S}}, \qquad \text{(Lemma 2.)}$$

$$= \left\|\mathbf{S}^{-1}(\mathbf{I}_m + \mathbf{D})^{-1}\mathbf{S}^{-\top}\mathbb{E}\left[X\xi^{\top}\right]\right\|_{\mathbf{S}^{\top}\mathbf{S}}, \qquad (\mathbf{S}, \mathbf{S}^{\top} \text{ invertible.})$$

$$= \left\|\mathbf{S}\mathbf{S}^{-1}(\mathbf{I}_m + \mathbf{D})^{-1}\mathbf{S}^{-\top}\mathbb{E}\left[X\xi^{\top}\right]\right\|, \qquad \text{(Switch to } \ell_2 \text{ norm.)}$$

$$= \left\|(\mathbf{I}_m + \mathbf{D})^{-1}\mathbf{S}^{-\top}\mathbb{E}\left[X\xi^{\top}\right]\right\|,$$

$$\leq \left\|\mathbf{S}^{-\top}\mathbb{E}\left[X\xi^{\top}\right]\right\|, \qquad (28)$$

$$= \left\|\mathbf{S}\mathbf{S}^{-1}\mathbf{S}^{-\top}\mathbb{E}\left[X\xi^{\top}\right]\right\|, \qquad \text{(Substitute in } \mathbf{I}_m = \mathbf{S}\mathbf{S}^{-1}.)$$

$$= \left\|\mathbf{S}^{-1}\mathbf{S}^{-\top}\mathbb{E}\left[X\xi^{\top}\right]\right\|_{\mathbf{S}^{\top}\mathbf{S}}, \qquad \text{(Back to weighted norm.)}$$

$$= \left\|\mathbb{E}\left[XX^{\top}\right]^{-1}\mathbb{E}\left[X\xi^{\top}\right]\right\|_{\boldsymbol{\Sigma}_X^{\mathfrak{A}}}, \qquad \text{(Substitute in } \boldsymbol{\Sigma}_X^{\mathfrak{A}} := \mathbb{E}^{\mathfrak{A}}\left[XX^{\top}\right] = \mathbf{S}^{\top}\mathbf{S}.)$$

$$= \left\|\mathbf{f} + \mathbb{E}\left[XX^{\top}\right]^{-1}\mathbb{E}\left[X\xi^{\top}\right] - \mathbf{f}\right\|_{\boldsymbol{\Sigma}_X^{\mathfrak{A}}}, \qquad \text{(Add and subtract } \mathbf{f}.)$$

$$= \left\|\mathbb{E}\left[XX^{\top}\right]^{-1}\left(\mathbb{E}\left[XX^{\top}\right]\mathbf{f} + \mathbb{E}\left[X\xi^{\top}\right]\right) - \mathbf{f}\right\|_{\boldsymbol{\Sigma}_X^{\mathfrak{A}}}, \qquad \text{(Use } \mathbf{I}_m = \mathbb{E}\left[XX^{\top}\right]^{-1}\mathbb{E}\left[XX^{\top}\right].)$$

$$= \left\|\mathbb{E}\left[XX^{\top}\right]^{-1}\mathbb{E}\left[X\left(\mathbf{f}^{\top}X + \xi\right)^{\top}\right] - \mathbf{f}\right\|_{\boldsymbol{\Sigma}_X^{\mathfrak{A}}}, \qquad \text{(Linearity of expectation.)}$$

$$= \left\|\mathbb{E}\left[XX^{\top}\right]^{-1}\mathbb{E}\left[XY^{\top}\right] - \mathbf{f}\right\|_{\boldsymbol{\Sigma}_X^{\mathfrak{A}}}, \qquad \text{(Structural eq. of } Y.)$$

$$= \left\|\hat{\mathbf{h}}_{\mathrm{ERM}}^{\mathfrak{A}} - \mathbf{f}\right\|_{\boldsymbol{\Sigma}_X^{\mathfrak{A}}}, \qquad \text{(ERM closed form solution.)}$$

where inequality Eq. (28) holds because $\mathbf{D}$ is non-negative diagonal. Furthermore, inequality Eq. (28) only holds with equality iff $\mathbf{S}^{-\top}\mathbb{E}\left[X\xi^{\top}\right]$ is in the kernel of $\mathbf{D}$. Or equivalently, iff $\mathbb{E}\left[X\xi^{\top}\right]$ is in the kernel of $\mathbf{S}^{\top}\mathbf{D}\mathbf{S} = \boldsymbol{\Sigma}_{\tilde{G}}$, which from Lemma 1 is true iff $\mathbb{E}^{\mathfrak{A}}[GX \mid G] \perp \mathbb{E}^{\mathfrak{A}}[X \mid \xi]$ a.s. □

### F.6 Miscellaneous supporting lemmas

**Lemma 1** (Gaussian conditional orthogonality lemma). *Let $X, Y, Z \in \mathbb{R}^n$ be zero-mean jointly Gaussian random vectors with covariance matrices $\boldsymbol{\Sigma}_X = \mathbb{E}[XX^\top]$, $\boldsymbol{\Sigma}_Z = \mathbb{E}[ZZ^\top]$, and cross-covariance $\boldsymbol{\Sigma}_{Y,Z} = \mathbb{E}[YZ^\top]$. Define the conditional expectation*

$$\mathbb{E}[Y \mid Z] := \left(\mathbb{E}\big[ZZ^\top\big]^{-1}\mathbb{E}\big[ZY^\top\big]\right)^\top Z = \boldsymbol{\Sigma}_{Y,Z}\boldsymbol{\Sigma}_Z^{-1}Z.$$

*Then the following are equivalent:*

$$X \perp \mathbb{E}[Y \mid Z] = 0 \quad a.s. \qquad \Longleftrightarrow \qquad \boldsymbol{\Sigma}_X\boldsymbol{\Sigma}_{Y,Z} = \mathbf{0}.$$

*Proof.* Since $X, Y, Z$ are jointly Gaussian, $\mathbb{E}[Y \mid Z] = \mathbf{M}Z$ with $\mathbf{M} := \boldsymbol{\Sigma}_{Y,Z}\boldsymbol{\Sigma}_Z^{-1}$. The scalar random variable

$$S := X^\top\mathbb{E}[Y \mid Z] = X^\top\mathbf{M}Z$$

is Gaussian with mean zero. Hence,

$$S = 0 \quad \text{a.s.} \qquad \Longleftrightarrow \qquad \mathrm{Var}(S) = 0.$$

Compute the variance:

$$\mathrm{Var}(S) = \mathbb{E}\big[S^2\big] = \mathbb{E}\big[(X^\top\mathbf{M}Z)^2\big] = \mathbb{E}\big[Z^\top\mathbf{M}^\top XX^\top\mathbf{M}Z\big].$$

Using independence and zero-mean assumptions,

$$\mathrm{Var}(S) = \mathrm{tr}\big(\mathbf{M}^\top\boldsymbol{\Sigma}_X\mathbf{M}\boldsymbol{\Sigma}_Z\big).$$

Since covariance matrices are positive semidefinite, $\mathrm{Var}(S) = 0$ iff

$$\boldsymbol{\Sigma}_X^{1/2}\mathbf{M}\boldsymbol{\Sigma}_Z^{1/2} = \mathbf{0} \implies \boldsymbol{\Sigma}_X\mathbf{M}\boldsymbol{\Sigma}_Z = \mathbf{0}.$$

Substituting $\mathbf{M} = \boldsymbol{\Sigma}_{Y,Z}\boldsymbol{\Sigma}_Z^{-1}$ gives

$$\boldsymbol{\Sigma}_X\boldsymbol{\Sigma}_{Y,Z} = \mathbf{0},$$

completing the proof. □

**Lemma 2** (SPD and PSD simultaneous denationalization via congruence). *For any $n \times n$ matrices $\mathbf{A} \succ \mathbf{0}$, $\mathbf{B} \succcurlyeq \mathbf{0}$, there exists an invertible $\mathbf{S} \in \mathbb{R}^{n \times n}$ and non-negative diagonal $\mathbf{D} \in \mathbb{R}^{n \times n}$ such that*

$$\mathbf{A} = \mathbf{S}^\top\mathbf{S}, \qquad\qquad \mathbf{B} = \mathbf{S}^\top\mathbf{D}\mathbf{S}.$$

*Proof.* This is similar to Theorem 7.6.4 in [79, p. 465] for two SPD matrices. We proceed similarly; Since $\mathbf{A}$ is SPD, it admits a unique SPD square root $\mathbf{A}^{1/2}$. Define

$$\mathbf{C} := \mathbf{A}^{-1/2}\mathbf{B}\mathbf{A}^{-1/2},$$

which is SPD. By the spectral theorem, there exists an orthogonal matrix $\mathbf{U}$ such that

$$\mathbf{C} = \mathbf{U}^\top\mathbf{D}\mathbf{U},$$

where $\mathbf{D}$ is diagonal with non-negative entries (the eigenvalues of $\mathbf{C}$). Set

$$\mathbf{S} := \mathbf{U}\mathbf{A}^{1/2}.$$

Then

$$\mathbf{S}^\top\mathbf{S} = \mathbf{A}^{1/2}\mathbf{U}^\top\mathbf{U}\mathbf{A}^{1/2} = \mathbf{A}^{1/2}\mathbf{I}\mathbf{A}^{1/2} = \mathbf{A},$$

and

$$\mathbf{S}^\top\mathbf{D}\mathbf{S} = \mathbf{A}^{1/2}\mathbf{U}^\top\mathbf{D}\mathbf{U}\mathbf{A}^{1/2} = \mathbf{A}^{1/2}\mathbf{C}\mathbf{A}^{1/2} = \mathbf{B}.$$

Since $\mathbf{A}^{1/2}$ and $\mathbf{U}$ are invertible, $\mathbf{S}$ is invertible, completing the proof. □

**Lemma 3** (solvability of simultaneous SEM). *The SEM $\mathfrak{M}$ in Example 1 is solvable iff $\mathbf{f}^\top \boldsymbol{\tau}^\top \neq 1$, in which case the following solution defines the reduced form of the SEM.*

$$\begin{bmatrix} X \\ Y \end{bmatrix} = \begin{bmatrix} \mathbf{I}_m & -\boldsymbol{\tau}^\top \\ -\mathbf{f}^\top & 1 \end{bmatrix}^{-1} \left( \begin{bmatrix} \boldsymbol{\Gamma}^\top \\ \mathbf{0}_{1 \times k} \end{bmatrix} Z + \begin{bmatrix} \mathbf{T}^\top \\ \boldsymbol{\epsilon}^\top \end{bmatrix} C + \sigma \cdot \begin{bmatrix} N_X \\ N_Y \end{bmatrix} \right),$$

*Similarly, SEM $\mathfrak{A}$ in Example 2 solves for $\mathbf{f}^\top \boldsymbol{\tau}^\top \neq \kappa^{-1}$.*

*Proof.* We re-state the SEM $\mathfrak{M}$ in the following block form

$$\begin{bmatrix} X \\ Y \end{bmatrix} = \begin{bmatrix} \mathbf{0}_{m \times m} & \boldsymbol{\tau}^\top \\ \mathbf{f}^\top & \mathbf{0}_{1 \times 1} \end{bmatrix} \begin{bmatrix} X \\ Y \end{bmatrix} + \begin{bmatrix} \boldsymbol{\Gamma}^\top \\ \mathbf{0}_{1 \times k} \end{bmatrix} Z + \begin{bmatrix} \mathbf{T}^\top \\ \boldsymbol{\epsilon}^\top \end{bmatrix} C + \sigma \cdot \begin{bmatrix} N_X \\ N_Y \end{bmatrix},$$

$$\Rightarrow \begin{bmatrix} \mathbf{I}_m & -\boldsymbol{\tau}^\top \\ -\mathbf{f}^\top & 1 \end{bmatrix} \cdot \begin{bmatrix} X \\ Y \end{bmatrix} = \begin{bmatrix} \boldsymbol{\Gamma}^\top \\ \mathbf{0}_{1 \times k} \end{bmatrix} Z + \begin{bmatrix} \mathbf{T}^\top \\ \boldsymbol{\epsilon}^\top \end{bmatrix} C + \sigma \cdot \begin{bmatrix} N_X \\ N_Y \end{bmatrix}$$

solving for $(X, Y)$ involves inverting the block matrix on the LHS. The result immediately follows from Proposition 2.8.7 in [80, p. 108], via the Schur complement formula for block matrix inversion. $\square$

