# OpenReview forum: "An Analysis of Causal Effect Estimation using Outcome Invariant Data Augmentation"
_NeurIPS.cc/2025/Workshop/Reliable_ML — NeurIPS 2025 - Reliable ML Workshop_

### Official Review · Reviewer_MTF6 · 2025-09-08
**Incorporating data augmentation as instruments**

**Rating:** 7
**Confidence:** 4

**Review:**

Summary: The author proposes a novel approach to incorporating data augmentation; they seek to use augmentation as instruments and incorporates them in a manner that resembles nonparametric IV regressions. They provide some basic justification in the case of linear models for why this approach works, and then illustrates improvements through this approach that are meaningful.

Strengths: Novel methodology, some basic justification, meaningful improvements on simulation and real data.

Weakness: The writing is quite poor and the focus is unclear. There's a lot on estimating the ATE, and as far as I'm aware that tangential to the goal of the paper. It can be reorganized de-emphasizing the content on causal effects since the goal really is not to estimate a causal effect but still to estimate some function. IMO, almost all content relating to causal graphs could probably be removed without losing much, and would probably enhance the readability.

Suggestions: Some literature is missing; for example the setting is a special case of the nonparametric IV setting described in Newey and Powell. Furthermore, although equation (5) is correct, the description of a procedure underneath it is invalid. This is known in the econometrics literature as the "forbidden regression" and known to be an inconsistent estimator; see Angrist and Pischke for example.

---

### Official Review · Reviewer_AZcs · 2025-09-22
**An Analysis of Causal Effect Estimation using Outcome Invariant Data Augmentation**

**Rating:** 8
**Confidence:** 2

**Review:**

# Summary
While some regularization methods have been used to reduce confounding bias, data augmentation (DA) does not have any such guarantees. Authors first relax the properties of instrumental variables to introduce IV-like (IVL regression) and show that this does not identify causal effects, but does reduce confounding bias. They then show that DA (given f is invariant) reduces confounding bias in causal effect estimation. In simulated and real-data experiments, they show that given confounding, estimation methods using DA reduce bias over others.

# Strengths
1. Authors present an extensive preliminary section introducing key aspects of causal analysis and data augmentation
2. Short discussion (lines 118-121) of the invariance assumption as it relates to image data was very useful to ground the assumption of invariant DAs
3. Inclusion of theoretical results on decrease in confounding bias as well as empirical simulations using both simulated and real world data

# Weaknesses/Limitations
1. The structure of the paper is somewhat atypical, which makes it harder than necessary to follow (see specific suggestions)

# Suggestions for Authors
1. The presentation of contributions could be made easier to follow by making it follow the structure of the paper. As it stands, it's a bit strange to see contribution (i) be Section 4.1, but (ii) is Section 3 and then (iii) is Section 4.2. It makes it bit hard to read the paper with the full understanding of what is coming up next.
2. Placing Related Work as section 5 seems out of place. Given the extensive preliminaries, it is not hugely needed earlier in the paper, but does not seem fitting between theory and empirical.